# MiT/TFE factors control ER-phagy via transcriptional regulation of FAM134B

Laura Cinque[1], Chiara De Leonibus[1], Maria Iavazzo[1], Natalie Krahmer[2,3], Daniela Intartaglia[1], Francesco Giuseppe Salierno[1], Rossella De Cegli[1], Chiara Di Malta[1], Maria Svelto[1], Carmela Lanzara[1], Marianna Maddaluno[1], Luca Giorgio Wanderlingh[1], Antje K Huebner[4], Marcella Cesana[1,5], Florian Bonn[6], Elena Polishchuk[1], Christian A Hübner[4], Ivan Conte[1,7] (iD), Ivan Dikic[4,8] (iD), Matthias Mann[9,10] (iD), Andrea Ballabio[1,11,12,13] (iD), Francesca Sacco[14], Paolo Grumati[1] & Carmine Settembre[1,11,*] (iD)

## Abstract

Lysosomal degradation of the endoplasmic reticulum (ER) via autophagy (ER-phagy) is emerging as a critical regulator of cell homeostasis and function. The recent identification of ER-phagy receptors has shed light on the molecular mechanisms underlining this process. However, the signaling pathways regulating ER-phagy in response to cellular needs are still largely unknown. We found that the nutrient responsive transcription factors TFEB and TFE3—master regulators of lysosomal biogenesis and autophagy—control ER-phagy by inducing the expression of the ER-phagy receptor FAM134B. The TFEB/TFE3-FAM134B axis promotes ER-phagy activation upon prolonged starvation. In addition, this pathway is activated in chondrocytes by FGF signaling, a critical regulator of skeletal growth. FGF signaling induces JNK-dependent proteasomal degradation of the insulin receptor substrate 1 (IRS1), which in turn inhibits the PI3K-PKB/Akt-mTORC1 pathway and promotes TFEB/TFE3 nuclear translocation and enhances FAM134B transcription. Notably, FAM134B is required for protein secretion in chondrocytes, and cartilage growth and bone mineralization in medaka fish. This study identifies a new signaling pathway that allows ER-phagy to respond to both metabolic and developmental cues.

**Keywords** ER-phagy; Fam134B; FGF signaling; IRS1/PI3K signaling; TFEB

**Subject Categories** Autophagy & Cell Death; Chromatin, Transcription & Genomics; Musculoskeletal System
**The EMBO Journal (2020) 39: e105696**

See also: **M Fraiberg & Z Elazar** (September 2020)

## Introduction

(Macro)autophagy is an evolutionarily conserved pathway devoted to the degradation of cytosolic components. Autophagy relies on the activity of two organelles, the autophagosomes (autophagic vesicles —AVs), which sequester substrates, and the lysosomes, where degradation occurs (Dikic & Elazar, 2018).

Initially described as non-specific, autophagy is now emerging as a selective process. Substrate selectivity in autophagy is facilitated by the so-called cargo receptors, which bind cargos and tether them to autophagosome membranes by direct interaction with LC3 or GABARAP proteins via LIR or GIM motifs, respectively (Kirkin & Rogov, 2019). Selective autophagy is particularly important for the maintenance of organelles in terms of quality and quantity (Okamoto, 2014).

1 Telethon Institute of Genetics and Medicine (TIGEM), Pozzuoli, Italy
2 Institute for Diabetes and Obesity, Helmholtz Zentrum München, Munich-Neuherberg, Germany
3 Department of Proteomics and Signal Transduction, Max Planck Institute of Biochemistry, Martinsried, Germany
4 Institute of Human Genetics, Jena University Hospital, Friedrich-Schiller-University Jena, Jena, Germany
5 Department of Advanced Biomedical Sciences, University of Naples "Federico II", Naples, Italy
6 Institute of Biochemistry II, Goethe University Frankfurt – Medical Faculty, University Hospital, Frankfurt am Main, Germany
7 Department of Biology, University of Naples "Federico II", Naples, Italy
8 Buchmann Institute for Molecular Life Sciences, Goethe University, Frankfurt, Frankfurt am Main, Germany
9 Department of Proteomics and Signal Transduction, Max Planck Institute of Biochemistry, Martinsried, Germany
10 Faculty of Health Sciences, NNF Center for Protein Research, University of Copenhagen, Copenhagen, Denmark
11 Department of Translational Medicine, Federico II University, Naples, Italy
12 Jan and Dan Duncan Neurological Research Institute, Texas Children Hospital, Houston, TX, USA
13 Department of Molecular and Human Genetics, Baylor College of Medicine, Houston, TX, USA
14 Department of Biology, University of Rome "Tor Vergata", Rome, Italy
 *Corresponding author. Tel: +39 081 19230601; E-mail: settembre@tigem.it

In particular, autophagy of the ER (ER-phagy) has been implicated in several ER functions, such as remodeling, response to starvation, and cargo-quality control (Grumati *et al*, 2018; De Leonibus *et al*, 2019; Fregno & Molinari, 2019; Wilkinson, 2019). The incorporation of ER fragments into AVs is facilitated by distinct ER-phagy receptors, namely Fam134b, Rtn3, Atl3, Tex264, Sec62, and Ccpg1 (Khaminets *et al*, 2015; Grumati *et al*, 2017; Smith *et al*, 2018; An *et al*, 2019; Chen *et al*, 2019; Chino *et al*, 2019). In addition, 200 high-confidence ER-phagy regulators have been recently identified through a genome-wide CRISPRi screen (Liang *et al*, 2020).

To date, however, the signaling events that regulate ER-phagy in response to cellular and environmental cues are largely uncharacterized.

The whole autophagy process is regulated by post-translational modifications such as phosphorylation and ubiquitination (Dikic & Elazar, 2018). In addition, the biogenesis of autophagosomes and lysosomes is coordinated at the transcriptional level by TFEB and TFE3, members of the MiTF/TFE transcription factor family, which bind to a conserved 10-base E-box-like palindromic sequences, known as Coordinated Lysosomal Expression and Regulation (*CLEAR*) sites, in the promoter of several lysosomal and autophagy genes (Sardiello *et al*, 2009; Settembre *et al*, 2011; Martina *et al*, 2014).

The activity of TFEB and TFE3 factors is negatively regulated by nutrient and growth factor-sensitive kinases (such as mTORC1, AKT, ERK2, and GSK3B) and positively regulated by the phosphatase calcineurin through the modulation of the phosphorylation status of multiple serine residues (Puertollano *et al*, 2018). In particular, phosphorylation of TFEB and TFE3 by the mTORC1 kinase inhibits their nuclear translocation; therefore, the inhibition of mTORC1, for example during starvation, promotes TFEB and TFE3 dephosphorylation, nuclear translocation, and activation of target genes (Martina *et al*, 2012; Roczniak-Ferguson *et al*, 2012; Settembre *et al*, 2012). MiTF/TFE factors can be activated in response to multiple stimuli, such as exercise, infection, inflammation, and lipid overload, suggesting that they might participate to multiple cell- and tissue-specific responses (Puertollano *et al*, 2018). However, whether they are able to regulate selective autophagy is currently unknown.

We recently demonstrated that fibroblast growth factor (FGF) signaling, a critical regulator of skeletogenesis (Ornitz & Marie, 2015), activates autophagy in chondrocytes (Cinque *et al*, 2015). Moreover, chondrocytes that display defects in the autophagy pathway have altered ER morphology and function (Cinque *et al*, 2015; Bartolomeo *et al*, 2017; Kang *et al*, 2017; Horigome *et al*, 2019). These observations led us to hypothesize that ER-phagy in chondrocytes is regulated by the FGF signaling.

In this work, we characterized the signaling cascade activated by FGF in chondrocytes and demonstrated that FGF induces lysosome biogenesis and ER-phagy through the activation of TFEB and TFE3 transcription factors. Mechanistically, we found that TFEB and TFE3 induce the expression of the ER-phagy receptor FAM134B, hence stimulating the delivery of ER fragments into newly formed lysosomes. Starvation-mediated TFEB and TFE3 activation also induces ER-phagy via FAM134B induction in multiple cell lines, demonstrating the generality of this pathway. In addition, this process appears to be physiologically relevant during skeletal development both in fish and in mouse.

# Results

## FGF signaling induces lysosome biogenesis and ER-phagy in chondrocytes

High-resolution mass spectrometry-based proteomic analysis in rat chondrocytes (RCS) showed that FGF18 stimulation significantly downregulated ER and ribosome proteins while upregulating lysosome category compared with vehicle-stimulated cells (Fig 1A; Tables EV1 and EV2). Consistently, a prolonged (12 h), but not short term (4 h), stimulation of RCS with FGF18 or FGF2 increased lysosome numbers, and their degradative function, as assessed by *I)* LysoTracker fluorescence intensity; *II)* degradation of the artificial substrate DQ-BSA; *III)* quantification of lysosome-associated membrane protein 1 (LAMP1)-positive vesicles; and *IV)* enzymatic activity of tested lysosomal hydrolases (β-glucuronidase and of hexosaminidase; Figs 1B–D and EV1A–D). To probe the nature of the substrates degraded within lysosomes upon FGF stimulation, we pharmacologically inhibited lysosomal degradation with the vATPase inhibitor bafilomycin A1 (BafA1). We observed a striking accumulation of ER membranes decorated with ribosomes, but not of mitochondria, in lysosomes of FGF18-treated chondrocytes (Fig 1E). We confirmed this observation by tracing a photostable dye that selectively accumulates in the ER (ER-Tracker; Fig 1F) and by using the ER autophagy tandem reporter (EATR) assay (Liang *et al*, 2018; Fig 1G). The delivery of ER portions into lysosomes upon FGF stimulation was mediated by ER-phagy, since it did not occur in RCS lacking the autophagy gene *Atg9* (ATG9A[KO]) as well as *Atg7* and *Beclin1* genes (ATG7[KO]; BCN1[KO]; Figs 1G and EV1E and F). Taken together, these data demonstrate that FGF induced lysosome biogenesis and ER-phagy in chondrocytes.

## FGF18 stimulation inhibits the PI3K signaling via IRS1 degradation

Next, we sought to characterize the signaling pathway mediating these effects. FGF ligands can bind to four receptor tyrosine kinases (FGFR1-4; Ornitz & Marie, 2015). To test which receptor was effectively involved in FGF18 signaling, we deleted each of the FGFRs, alone or in combination, in RCS (Fig EV1G). FGF18 induction of lysosome biogenesis was lost in RCS lacking FGFR3 (FGFR3[KO]) and partially inhibited in cells lacking FGFR4 (FGFR4[KO]), while it remained unmodified in cells lacking FGFR1 and/or FGFR2 receptors (Figs 1H and EV1H). Notably, FGFR3/4[KO] chondrocytes showed lysosomal dysfunction, reduced lysosomal enzymatic activity (Fig EV1D and G–J), and abolished ER-phagy compared with control cells (Fig 1I and J). Thus, FGF18 promotes ER-phagy and lysosome biogenesis through FGFR3 and FGFR4.

Next, to characterize the signaling pathway downstream FGFR3/4, we performed quantitative- and phospho-proteomics and transcriptome analysis in control and FGFR3/4[KO] RCS stimulated with FGF18.

Phospho-proteomics allowed us to quantify the dynamic response of approximately 20,000 phospho-sites regulated by FGF signaling in RCS. We observed that insulin and PI3K signaling were the most significantly inhibited pathways upon FGF18 stimulation in control but not in FGFR3/4[KO] RCS (Fig 2A; Table EV3). Western blot analysis demonstrated that the activities of AKT and mTORC1

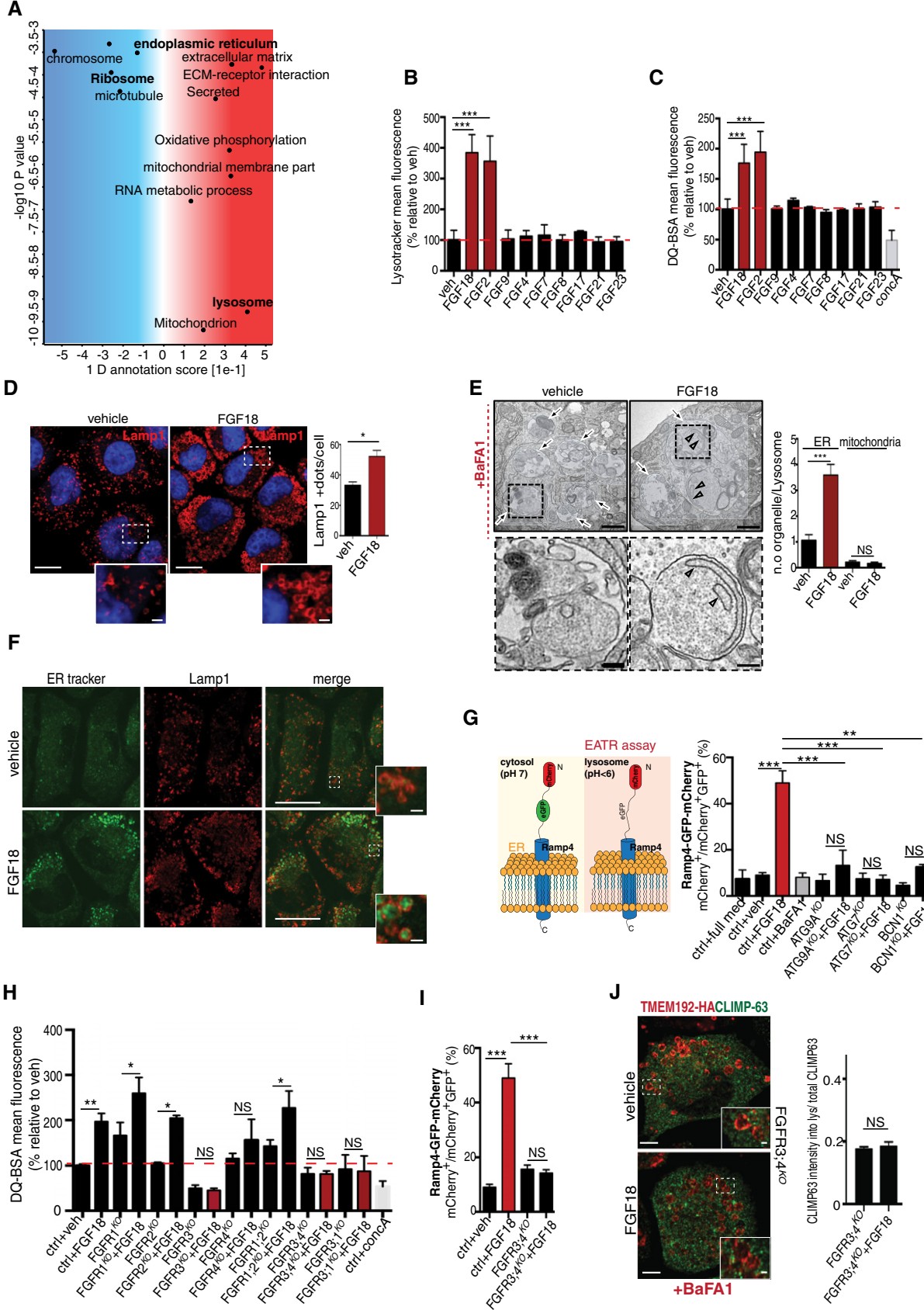

Figure 1.

**Figure 1.  FGF signaling promotes lysosome biogenesis and ER-phagy in chondrocytes through FGFR3 and FGFR4.**

A    MS proteomic analysis of RCS chondrocytes treated with vehicle (5% ABS) and FGF18 (50 ng/ml) for 16 h. Biological processes and cellular components regulated by FGF signaling are shown. FDR < 0.05. N = 4 biological replicates/treatment were analyzed. P-value was calculated using 1D annotation enrichment test based on Wilcoxon–Mann–Whitney test.

B, C  FACS analysis of LysoTracker (B) and DQ-BSA (C) dye fluorescence in RCS treated with the indicated FGF ligands (50 ng/ml) for 16 h. Concanamycin A was used at 100 nM for 1 h to inhibit lysosomal function. Fluorescence intensities were expressed as % relative to vehicle (5% ABS). Mean ± standard error of the mean (SEM) of N = 3 biological replicates/treatment. One-way analysis of variance (ANOVA) P < 0.002 (B) and P = 0.005 (C); Tukey's post hoc test ***P < 0.0005.

D    Lamp1 immunofluorescence (red) in RCS chondrocytes treated with vehicle (5% ABS) and FGF18 (50 ng/ml) for 16 h. Insets show magnification of the boxed area. Nuclei were stained with DAPI (blue). Scale bar 15 and 2 μm (higher magnification boxes). Bar graph shows quantification of Lamp1-positive vesicles/cell. Mean ± standard error of the mean (SEM) of N = 3 biological replicates/treatment. n = 45 cells were analyzed. Student's paired t-test *P < 0.05.

E    Representative TEM images of RCS chondrocytes treated with 5% ABS (vehicle) and FGF18 (50 ng/ml) for 16 h. BafA1 (100 nM; 4 h) was used to inhibit lysosome activity. Arrows indicate lysosomes. Higher magnification insets showed the presence of ER membranes decorated with ribosomes (arrowheads). Scale bar 500 nm. Quantification shows average number of ER membranes and mitochondria number/lysosome vesicle (Lys). Mean ± standard error of the mean (SEM) of N = 3 biological replicates/treatment. ER fragment/vesicle n = 60 (vehicle) and n = 72 (FGF18) cells were analyzed; mitochondria number/vesicle: n = 40 (vehicle) and n = 72 (FGF18) cells were analyzed. Student's paired t-test ***P < 0.0005; NS, not significant.

F    Representative co-immunofluorescence staining of ER (ER-Tracker BODIPY Green) and lysosomes (Lamp1, red) in RCS chondrocytes treated with vehicle (5% ABS) and FGF18 (50 ng/ml) for 16 h. Insets show higher magnification of boxed area. Scale bar 15 and 2 μm (higher magnification boxes). N = 3 biological replicates/treatment.

G    Schematic representation of EATR assay: eGFP fluorescence, but not mCherry, is lost at acidic pH. Chondrocytes with indicated genotypes (ctrl = wild type) were treated with FGF18 (50 ng/ml; 12 h) and BafA1 (200 nM; 3 h) where indicated. ER acidification was measured by FACS. Mean ± standard error of the mean (SEM) of N = 3 biological replicates. One-way analysis of variance (ANOVA) P < 0.0001; Tukey's post hoc test ***P < 0.0005; **P < 0.005; NS, not significant.

H    FACS analysis of DQ-BSA dye fluorescence in chondrocytes with indicated genotypes (ctrl = wild type) treated with FGF18 (50 ng/ml) for 16 h. Concanamycin A was used at 100 nM for 1 h to inhibit lysosomal degradation. Fluorescence intensities were expressed as % relative to vehicle (5% ABS). Mean ± standard error of the mean (SEM) of N = 3 biological replicates/treatment/genotype. Analysis of variance (ANOVA) P = 0.00279; Tukey's post hoc test **P < 0.005; *P < 0.05; NS, not significant.

I    EATR assay. Chondrocytes with indicated genotypes (ctrl = wild type) treated with FGF18 (50 ng/ml; 16 h) where indicated. ER acidification was measured by FACS. Mean ± standard error of the mean (SEM) of N = 17 (veh), N = 17 (FGF18), N = 4 (FGFR3;4$^{KO}$); N = 4 (FGFR3;4$^{KO}$ FGF18) biological replicates. One-way analysis of variance (ANOVA) P < 0.0001; Tukey's post hoc test ***P < 0.0005; NS, not significant.

J    Representative immunofluorescence staining of ER (CLIMP63, green) and lysosomes (TMEM192-HA, red) in FGFR3;4$^{KO}$ chondrocytes treated with vehicle (5% ABS) and FGF18 ( 50 ng/ml for 16 h). BafA1 was used at 100 nM for 4 h to inhibit lysosomal degradation. Quantification of relative CLIMP63 fluorescence into TMEM192-HA-positive vesicles. N = 3 biological replicates; n = 21 cells were analyzed. Mean ± standard error of the mean (SEM). Paired Student's t-test NS = not significant. Scale bar 10 and 1.3 μm (higher magnification boxes).

kinases, downstream of the insulin/PI3K signaling, were strongly reduced following FGF18 treatment (Fig EV2A). By mapping the phospho-proteome and proteome profiles on a literature-curated insulin/IGF signaling network, we identified the insulin receptor substrate 1 (IRS1) protein, the adaptor protein that transmits signals from the insulin/IGF receptors (Haeusler *et al*, 2018), as the most downregulated protein by FGF18 treatment in RCS (Fig 2B). The downregulation of IRS1 upon FGF18 stimulation was due to an increase in its degradation and prevented by treatment with the proteasome inhibitor MG132 (Fig 2C). The proteasomal degradation of IRS1 can be triggered by phosphorylation mediated by different kinases, such as JNK1/2, mTORC1, or S6K (Haeusler *et al*, 2018). FGF18 activated JNK1/2, as demonstrated by phosphorylation of c-JUN (Fig 2D). When JNK1/2 kinases were pharmacologically inhibited, IRS1 was no longer phosphorylated at serine 307 and degraded upon FGF18 treatment (Figs 2D and EV2B). Notably, IRS1 overexpression as well as JNK1/2 suppression rescued mTORC1 and AKT signaling inhibition in RCS treated with FGF18 (Fig 2D and E). These data demonstrate that FGF inhibits the insulin/IGF signaling pathway via JNK-mediated IRS1 degradation in RCS.

## FGF signaling induces ER-phagy via FAM134B

QuantSeq 3′ mRNA sequencing gene expression analysis (GSE120516) followed by GOEA, with the output restricted to cellular compartment terms (CC_FAT), and KEGG pathway analysis identified lysosome as the most upregulated category by FGF signaling (Fig EV2C; Tables EV4–EV6), suggesting that transcriptional mechanisms might mediate the effects of FGF on lysosome

biogenesis and ER-phagy. Consistently, chondrocytes treated with the transcriptional inhibitor actinomycin failed to induce ER-phagy and lysosome biogenesis upon FGF18 stimulation (Fig 3A–C). A manually curated analysis of this list detected 66 lysosomal and autophagy genes significantly induced by FGF18 in WT but not in FGFR3/4$^{KO}$ RCS (Fig 3D; Tables EV4–EV6). In addition, among all ER-phagy receptors, FGF18 stimulation significantly increased mRNA levels of the *Fam134b* gene, which was measured using DNA primers located in exon 9 that is shared by all *Fam134b* isoforms (Figs 3E and EV2D). Fam134b protein was also induced by FGF18 (Fig 3F). FGF18 stimulation had moderate or null effects on other members of the FAM134 family, namely *Fam134a* and *Fam134c,* and on other ER-phagy receptors (*Ccpg1, Atl3, Sec62, Tex264, Rtn3*; Fig 3E). CRISPR/Cas9-mediated deletion of the C-terminal region of Fam134b, which contains the LIR domain required for the incorporation of the ER into autophagosomes (Khaminets *et al*, 2015; Fam134b ΔLIR; Fig 3F), significantly inhibited ER-phagy upon FGF18 stimulation (Fig 3G and H). These data suggest that FGF signaling induced lysosome biogenesis and Fam134b-mediated ER-phagy at least in part through transcriptional mechanisms.

## FGF signaling and nutrient starvation activate ER-phagy via TFEB/TFE3-mediated FAM134B induction

We postulated that FGF suppresses IRS1-PI3K downstream signaling and in turn activates a transcriptional program that promotes Fam134b-mediated ER-phagy and lysosome biogenesis in RCS. The kinases downstream IRS1 signaling, such as mTORC1 and AKT,

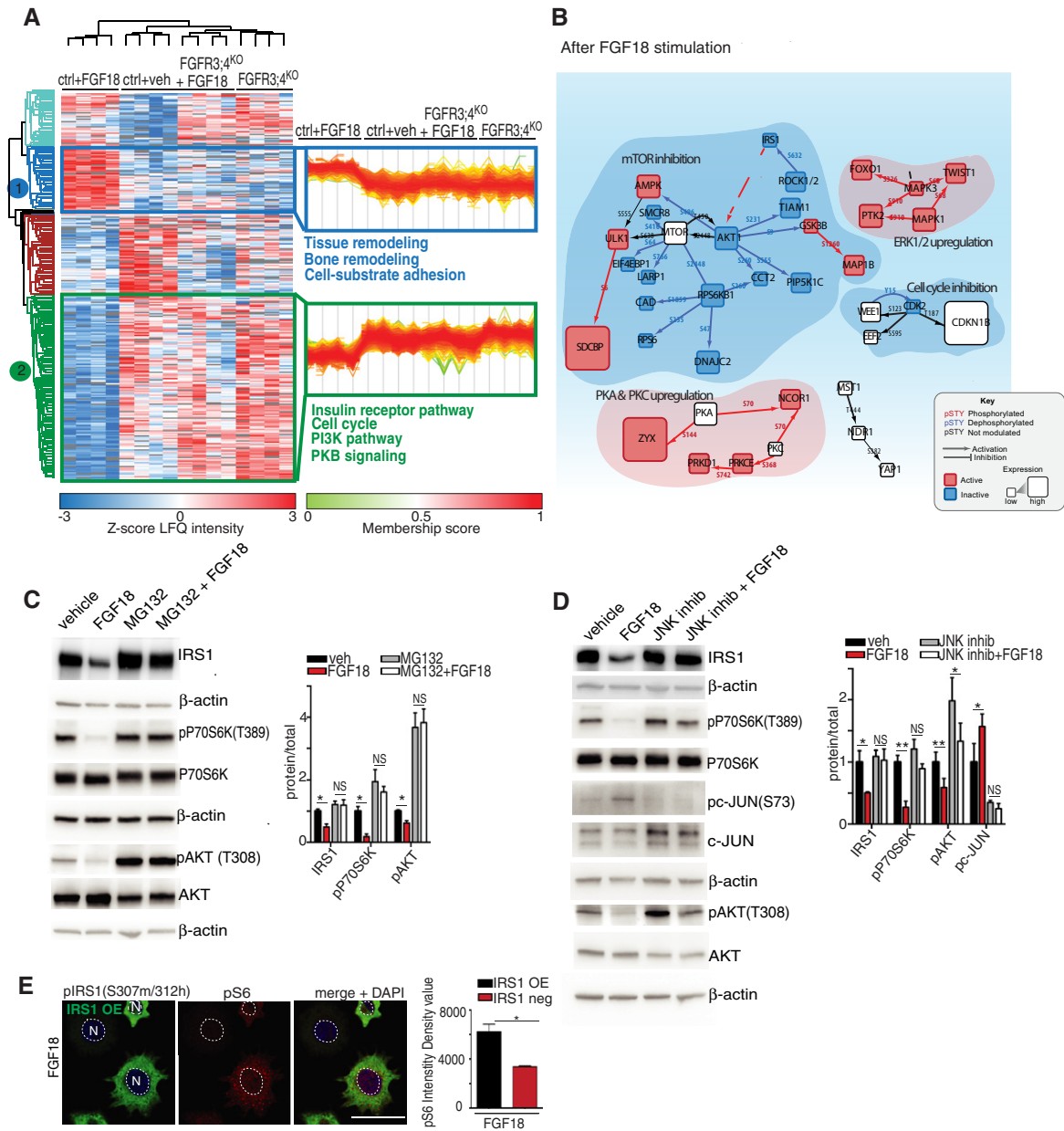

**Figure 2. FGF signaling inhibits the insulin/IGF signaling via JNK-mediated IRS1 degradation.**

A    MS phospho-proteomics analysis of RCS with indicated genotypes (ctrl = wild type) treated with vehicle (5% ABS) and FGF18 (50 ng/ml) for 16 h, showing biological processes regulated by FGF signaling (in blue: upregulated, in green: downregulated). N = 4 biological replicates were analyzed. FDR < 0.05.

B    Proteomic/phospho-proteomic signaling network modulated by FGF18 in chondrocytes. Red = activating phosphorylation; blue = inhibitory phosphorylation. Cube dimensions and colors are relative to level of protein regulations.

C, D    Representative Western blot analysis of IRS1, p-P70S6K (T389), P70S6K, p-AKT (T308), AKT, p-c-JUN (s73), and c-JUN in RCS chondrocytes treated with vehicle (5% ABS) and FGF18 (50 ng/ml) for 12 h. MG132 (10 μM for 6 h) and JNK inhibitor (50 μM for 12 h) to inhibit proteasome (C) and JNK (D) activity, respectively. β-actin was used as a loading control. N = 3 independent experiments. Bar graphs showed quantification of indicated proteins normalized to their totals. IRS1 was normalized to β-actin. Mean ± standard error of the mean (SEM). Student's paired t-test **P < 0.005; *P < 0.05; NS, not significant.

E    Co-immunofluorescence of p-IRS1 S307 mouse (green) and p-S6 S240/S242 (red) ribosomal protein in IRS1-overexpressing RCS chondrocytes treated with FGF18 (50 ng/ml) for 12 h. Nuclei (N) were stained with DAPI (blue). Scale bar 15 μm. Quantification analysis of p-S6 fluorescence intensity in IRS1-overexpressing vs non-expressing RCS chondrocytes. Mean ± standard error of the mean (SEM) of N = 3 biological replicates. n = 35 cells were analyzed. Student's paired t-test *P < 0.05.

Source data are available online for this figure.

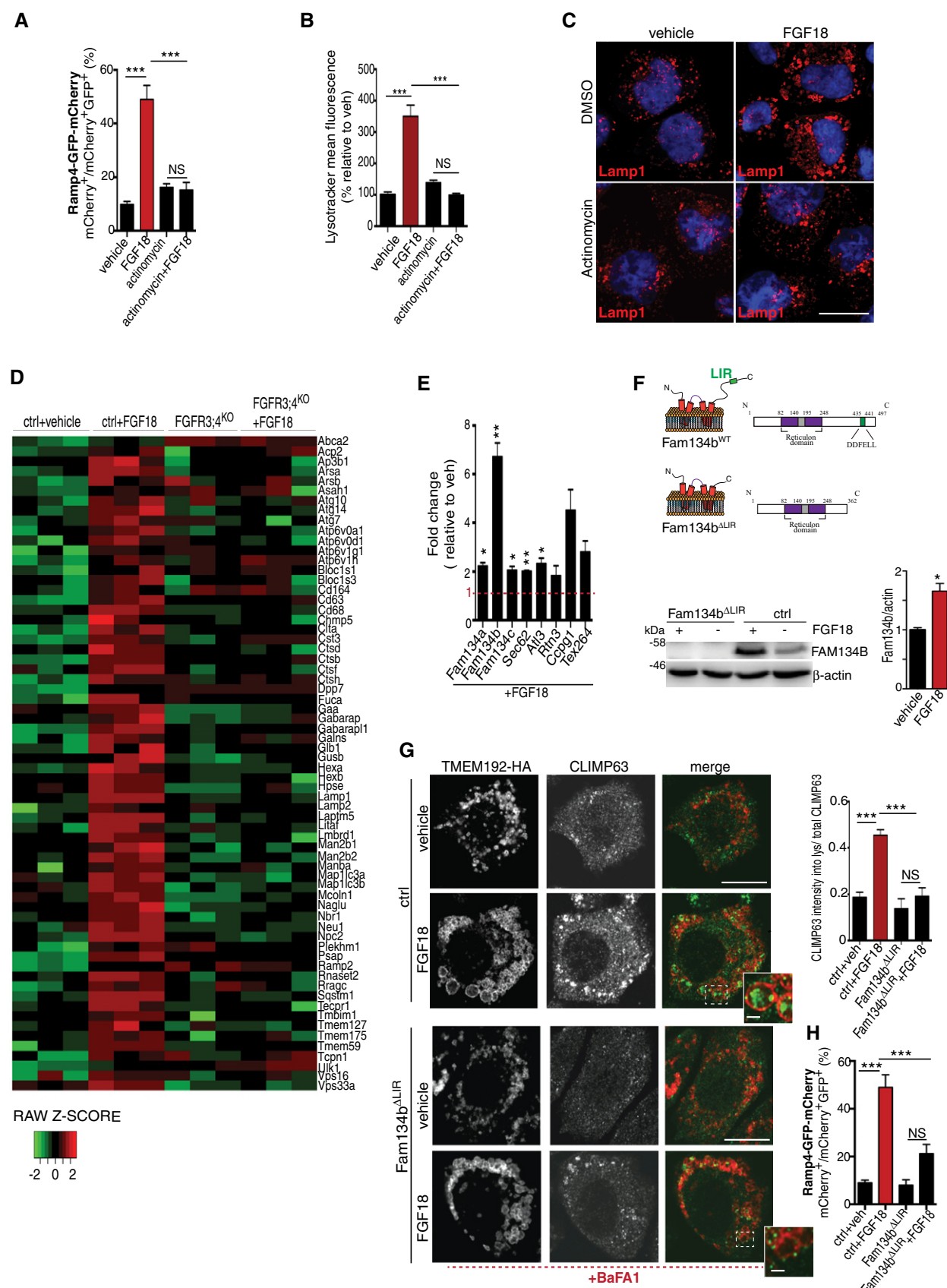

**Figure 3.**

**Figure 3. FGF signaling induces ER-phagy via FAM134B.**

A  EATR assay. Chondrocytes were treated with FGF18 (50 ng/ml; 16 h) and actinomycin (1 μg/ml; last 4 h) where indicated. ER acidification was measured by FACS. Mean ± standard error of the mean (SEM) of $N = 17$ (veh), $N = 17$ (FGF18), $N = 5$ (actinomycin), and $N = 5$ (actinomycin + FGF18) biological replicates. One-way analysis of variance (ANOVA) $P < 0.0001$; Tukey's post hoc test ***$P < 0.0005$; NS, not significant.

B  FACS analysis of LysoTracker dye fluorescence in RCS chondrocytes treated with FGF18 (50 ng/ml) for 16 h. Actinomycin (1 μg/ml) was added for the last 4 h. Fluorescence intensities were calculated as % relative to vehicle (5% ABS). Mean ± standard error of the mean (SEM) of $N = 3$ biological replicates. One-way analysis of variance (ANOVA) $P = 0.0002$; Tukey's post hoc test ***$P < 0.0005$; NS, not significant.

C  Representative immunofluorescence staining of Lamp1 (red) in RCS chondrocytes treated with vehicle (5% ABS), FGF18 (50 ng/ml; 16 h) and actinomycin (1 μg/ml; 4 h). Nuclei were stained with DAPI (blue). Representative images of $N = 3$ biological replicates. Scale bar 10 μm.

D  Heatmap of lysosomal and autophagy gene expression in RCS chondrocytes with indicated genotypes (ctrl = wild type), treated with vehicle (5% ABS) and FGF18 (50 ng/ml) for 16 h. FDR < 0.05. $N = 3$ biological replicates/treatment were analyzed. In green: downregulated; in red: upregulated gene expression.

E  qRT–PCR analysis of ER-phagy receptors in RCS chondrocytes. Gene expression was analyzed after FGF18 (50 ng/ml) treatment for 16 h. Fold change values were relative to vehicle (5% ABS) and normalized to *Cyclophilin* gene. Mean ± standard error of the mean (SEM) of $N = 3$ biological replicates/treatment. Student's paired $t$-test **$P < 0.005$; *$P < 0.05$.

F  (top) Representative model of Fam134bΔLIR protein. LIR = LC3-interacting region. (bottom) Western blot analysis of Fam134b in chondrocytes with indicated genotypes (ctrl = wild type) treated with FGF18 (50 ng/ml) for 16 h. Representative images of $N = 3$ biological replicates/treatment. β-actin was used as a loading control. Bar graph showed quantification of Fam134b band intensity normalized to β-actin. Mean ± standard error of the mean (SEM). Student's paired $t$-test, *$P < 0.05$.

G  Immunofluorescence staining of CLIMP63 (green) and lysosomes (TMEM192-HA, red) in RCS chondrocytes with indicated genotypes (ctrl = wild type) upon FGF18 treatment (50 ng/ml for 16 h). BafA1 (100 nM; 4 h) was used to inhibit lysosomal degradation; scale bar 20 μm. Insets show magnification of CLIMP63 accumulation into lysosomes; scale bar 2 μm. Quantification of CLIMP63 fluorescence intensity into TMEM192-HA decorated lysosomes. Mean ± standard error of the mean (SEM) of $N = 3$ biological replicates. $n = 10$ cells/experiment were analyzed. One-way analysis of variance (ANOVA) $P = 1.55e^{-7}$; Tukey's post hoc test ***$P < 0.0005$; NS, not significant.

H  EATR assay in RCS with indicated genotypes (ctrl = wild type) showing % of cells with acidified ER by FACS analysis. FGF18 was used at 50 ng/ml for 16 h. Mean ± standard error of the mean (SEM) of $N = 4$ biological replicates/treatment/genotype. One-way analysis of variance (ANOVA) $P < 0.0001$; Tukey's post hoc test ***$P < 0.0005$; NS, not significant.

Source data are available online for this figure.

phosphorylate and inhibit TFEB and TFE3, members of the microphthalmia-associated transcription factor (MiTF/TFE) family (Hemesath *et al*, 1994), which regulates lysosome biogenesis and autophagy (Sardiello *et al*, 2009; Settembre *et al*, 2011; Martina *et al*, 2014; Puertollano *et al*, 2018). Notably, 55 out of 66 lysosomal/autophagy genes induced by FGF18 presented one or more TFEB-binding site (CLEAR; Sardiello *et al*, 2009) in their promoters (Table EV6), suggesting the testable hypothesis that TFEB and TFE3

mediate lysosome biogenesis by FGF. Consistently, FGF18 stimulation induced TFEB and TFE3 dephosphorylation and nuclear translocation in control but not in FGFR3/4$^{KO}$ RCS (Figs 4A and B, and EV2E). Furthermore, FGF18 promoted TFEB transcriptional activation, as demonstrated by an increased binding to the promoter of the target gene *Mucolipin-1* (Fig EV2F). The induction of lysosomal gene expression and lysosome degradative functions by FGF stimulation was significantly impaired in TFEB and TFE3 double knock-

**Figure 4. FGF induces nuclear translocation of TFEB/TFE3 transcription factors through JNK-mediated IRS1 degradation.**

A  TFE3 (green) subcellular localization in RCS with indicated genotypes (control = wild type) treated with FGF18 (50 ng/ml) for 16 h. Torin 1 (1 μM for 2 h) was used as positive control. Nuclei were stained with DAPI (blue). Bar graph shows quantification (expressed as %) of cells with nuclear TFE3 and TFEB (representative images of TFEB immunofluorescence are shown in Fig EV2E). Mean ± standard error of the mean (SEM) of $N = 3$ biological replicates; $n = 80$ (TFE3) and $n = 70$ (TFEB) cells/experiment were analyzed. Scale bar 10 μm. One-way analysis of variance (ANOVA) $P = 3.23e^{-12}$ (TFEB), $P = 2.48e^{-11}$ (TFE3); Tukey's post hoc test ***$P < 0.0005$; NS, not significant.

B  Western blot analysis of TFEB, and phospho-TFEB (Serine 142) in RCS chondrocytes stably expressing human TFEB-3XFlag protein. Cells were treated with vehicle (5% ABS) or FGF18 (50 ng/ml) for 16 h. β-actin was used as a loading control. Representative images of three independent experiments.

C  GFP immuno-EM staining in FGFR3/4$^{KO}$ chondrocytes infected with a constitutive nuclear (and active) mutant TFEB fused to GFP tag (TFEB- S142A:S211A-GFP). GFP-positive gold immune particles showed the presence of GFP puncta into the nucleus (*N*, stained in green for visualization). Lysosomes were stained in blue for visualization. Insets show magnification of lysosomes. Quantification of lysosome diameter (nm) in control (wild type) and FGFR3;4$^{KO}$ chondrocytes infected with empty or with TFEB- S142A:S211A-GFP vector. Mean ± standard error of the mean (SEM) of $N = 3$ biological replicates. $n = 40$ (veh), $n = 78$ (FGFR3;4$^{KO}$), and $n = 57$ (FGFR3;4$^{KO}$ +TFEB-S142A:S211A-GFP) cells were analyzed. Scale bar 500 nm. Kruskal–Wallis test $P = 1.43e^{-13}$; Nemenyi post hoc test ***$P < 0.0005$; NS, not significant.

D  Co-immunofluorescence of IRS1 and TFEB in IRS1-overexpressing RCS chondrocytes treated with FGF18 (50 ng/ml) for 12 h. Nuclei (*N*) were stained with DAPI (blue) and delimited with dashed line. Scale bar 15 μm. Quantification of TFEB nuclear localization in IRS1-overexpressing *vs* non-expressing RCS. Mean ± standard error of the mean (SEM) of $N = 3$ biological replicates. $n = 124$ cells were analyzed; Student's unpaired $t$-test *$P < 0.05$.

E  Subcellular localization analysis of TFEB (red) and TFE3 (green) in RCS chondrocytes treated with FGF18 (50 ng/ml) for 12 h. JNK inhibitor was used at 50 μM for 12 h. Nuclei were stained with DAPI (blue). Quantification analysis showed % of cells with nuclear TFEB and TFE3 in RCS chondrocytes with indicated treatments. Scale bar 15 μm. Mean ± standard error of the mean (SEM) of $N = 3$ biological replicates. $n = 126$ cells (control), 126 cells (FGF18), 95 cells (JNK inhibitor), 163 cells (JNK inhibitor + FGF18). Student's paired $t$-test *$P < 0.05$.

F  Western blot analysis of phospho-TFEB (S142) and IRS1 in RCS chondrocytes treated with vehicle (5% ABS) and FGF18 (50 ng/ml) for 12 h. JNK inhibitor was used at 50 μM for 12 h to inhibit kinase activity. β-actin was used as a loading control. Representative images of $N = 3$ independent experiments. Bar graphs show quantification of indicated proteins normalized to β-actin. Mean ± standard error of the mean (SEM). Student's paired $t$-test *$P < 0.05$; NS, not significant.

Source data are available online for this figure.

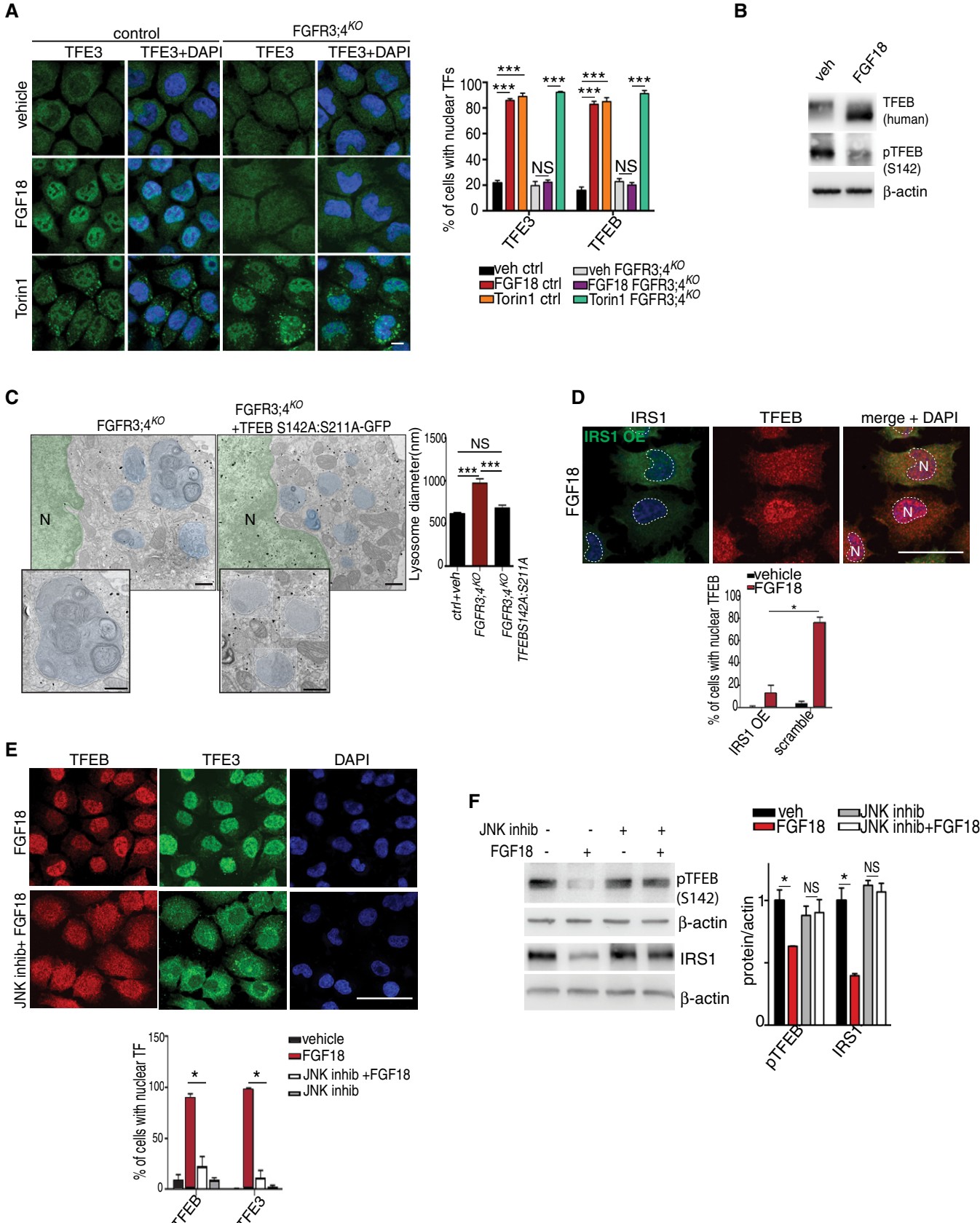

Figure 4.

out RCS (TFEB/3$^{KO}$; Fig EV2G–I), while forced overexpression of a constitutively active (nuclear) form of TFEB (TFEB S142A, S211A), or TFE3 (TFE3 S246A, S321A), rescued lysosomal gene expression

and lysosomal dysfunction in FGFR3/4$^{KO}$ RCS (Figs 4C and EV2J and K). Most importantly, TFEB and TFE3 activation by FGF18 was inhibited in RCS that overexpress IRS1, which are treated with JNK

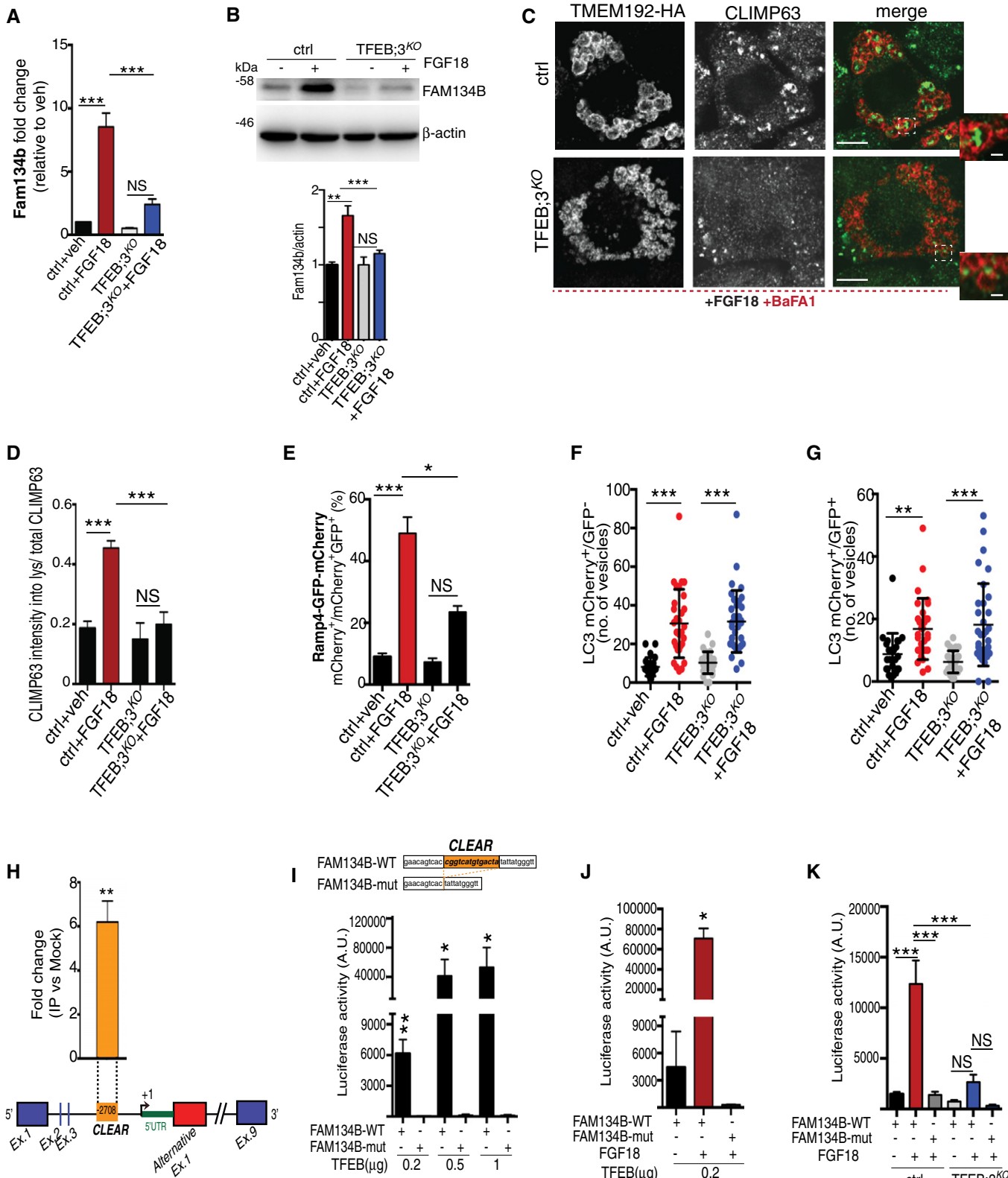

Figure 5.

**Figure 5.  FGF signaling regulates FAM134B transcriptional levels through TFEB and TFE3.**

A    qRT–PCR analysis of *Fam134b* gene expression in chondrocytes with indicated genotypes (ctrl = wild type) treated with vehicle (5% ABS) or with FGF18 (50 ng/ml; 16 h). Fold change values were relative to vehicle and normalized to *Cyclophilin* gene. Mean ± standard error of the mean (SEM) of $N = 3$ biological replicates. One-way analysis of variance (ANOVA) $P < 0.0001$: Tukey's post hoc test ***$P < 0.0005$; NS, not significant.

B    Western blot analysis of Fam134b protein in chondrocytes with indicated genotypes treated with vehicle (5% ABS) and FGF18 (50 ng/ml) overnight. β-actin was used as a loading control. Representative image of $N = 3$ biological replicates. Bar graph showed quantification of Fam134b normalized to β-actin. One-way analysis of variance (ANOVA) $P < 0.0001$; Sidak's multiple comparison test ***$P < 0.0005$; **$P < 0.005$ NS, not significant.

C    Co-staining of CLIMP63 (green) and TMEM192-HA (lysosomes, red) in control and TFEB;3$^{KO}$ RCS treated with FGF18 (50 ng/ml for 16 h). BaFA1 was used at 100 nM for 3 h. Scale bar 15 and 2 µm (higher magnification boxes).

D    Quantification of CLIMP63 fluorescence intensity into TMEM192-HA decorated lysosomes. Mean ± standard error of the mean (SEM) of $N = 3$ biological replicates/treatment/genotype (ctrl = wild type). $n = 10$ cells/experiment were analyzed. One-way analysis of variance (ANOVA) $P = 1.55e^{-7}$; Tukey's post hoc test ***$P < 0.0005$; NS, not significant.

E    EATR assay in chondrocytes with indicated genotypes (ctrl = wild type) showing % of cells with acidified ER measured by FACS. FGF18 was used at 50 ng/ml overnight. Mean ± standard error of the mean (SEM) of $N = 4$ biological replicates. One-way analysis of variance (ANOVA) $P < 0.0001$: Tukey's post hoc test ***$P < 0.0005$; *$P < 0.05$; NS, not significant.

F, G   Data plots show quantification of mCherry$^+$ vesicles/cell (autolysosomes) (F) and mCherry$^+$/GFP$^+$ vesicles/cell (autophagosomes) (G) in wild type (ctrl) and TFEB;3KO cells treated with vehicle (veh) or FGF18. $N = 3$ independent experiments. Mean ± standard error of the mean (SEM) of $N = 24$ (wild type treated with 5%ABS, veh), $N = 30$ (wild type treated with FGF18), $N = 27$ (TFEB;3KO veh), $N = 33$ (TFEB;3KO FGF18) cells. Student's unpaired *t*-test ***$P < 0.0005$; **$P < 0.005$.

H    ChIP analysis of TFEB binding to Fam134b DNA in RCS cells transfected with TFEB-3XFLAG. Numbers in the CLEAR site (yellow box) refer to the distance [in base pairs] from the transcriptional start site (+1) of *Fam134b-2* gene. Immunoprecipitated DNA was normalized to the input and plotted as relative enrichment over a mock control. Bar graph shows fold change enrichment; mean ± standard error of the mean (SEM) of $N = 3$ independent experiments. Student's unpaired *t*-test **$P < 0.005$.

I, J   Luciferase assays in RCS chondrocytes using as promoter a 0.7 kb genomic Fam134b DNA fragment containing a wild type (FAM134B-WT) or a deleted (FAM134B-mut) version of the CLEAR site. TFEB plasmid transfection amount and FGF18 (50 ng/ml for 16 h) treatments are indicated. Mean ± standard error of the mean (SEM) of $N = 3$ biological replicates. Student's paired *t*-test *$P < 0.05$; **$P < 0.005$.

K    Luciferase activity in wild type (ctrl) and TFEB;3$^{KO}$ RCS chondrocytes with indicated genotypes expressing the indicated Fam134b luciferase report plasmids and treated with FGF18 overnight (50 ng/ml) where indicated. Mean ± standard error of the mean (SEM) of $N = 5$ biological replicates. One-way analysis of variance (ANOVA) $P < 0.0001$: Sidak's multiple comparison test ***$P < 0.0005$; NS, not significant.

Source data are available online for this figure.

inhibitor or in which mTORC1 was constitutively activated by the overexpression of active mutant RagA/C-GTPases (Sancak *et al*, 2010; Figs 4D–F and EV3A). Taken together, these data indicate that FGF signaling induces TFEB/TFE3 activation through the suppression of the Ins/IGF1 signaling pathway in chondrocytes.

We tested whether TFEB and TFE3 also regulate ER-phagy in chondrocytes. The increase in Fam134b by FGF18 stimulation was blunted, at both mRNA and protein levels, in TFEB/3$^{KO}$ RCS (Fig 5A and B), and as a consequence, ER-phagy induction upon FGF18 stimulation was significantly impaired in TFEB/3$^{KO}$ RCS (Fig 5C–E). Notably, FGF18 was still inducing autophagy flux in TFEB/3$^{KO}$ cells (Figs 5F and G, and EV3B–D), suggesting that TFEB/TFE3 activation by FGF18 is essential for ER-phagy. The overexpression of TFEB, TFE3, and MITF, but not of FOXO3 or RUNX2, significantly increased mRNA and protein levels of Fam134b in chondrocytes (Fig EV3E and F), and this response showed a synergistic effect with FGF18 stimulation (Fig EV3E and F). Thus, FGF signaling induces ER-phagy via TFEB/TFE3-mediated Fam134b induction.

Chromatin immunoprecipitation (ChIP) and luciferase assay experiments demonstrated that TFEB binds to a CLEAR site that is located in the third intron of the *Fam134b* gene, in correspondence to the alternative transcript start site of the *Fam134b* isoform 2 (Fig 5H). FGF18 failed to promote luciferase expression in TFEB/3$^{KO}$ cells and in cells expressing a CLEAR-mutated version of the FAM134B promoter (Fig 5I–K). Consistently, FGF18 treatment predominantly upregulated the *Fam134b-2* isoform in RCS, largely in a TFEB/TFE3-dependent manner (Fig EV3G). Both Fam134b-1 and Fam134b-2 rescued ER-phagy defects in Fam134b ΔLIR cells (Fig EV3H and I). Taken together, these data indicate that Fam134b is a newly identified target gene of MiTF/TFE factors.

Next, we assessed the generalizability of ER-phagy regulation by the TFEB/TFE3-FAM134B axis. When RCS cells were starved in HBSS media, we observed TFEB dephosphorylation, nuclear translocation (Fig EV4A and B), *Fam134b-2* transcriptional induction (Fig EV4C), and ER-phagy activation (Fig EV4D and E). Notably, ER-phagy induction by HBSS starvation was significantly impaired in TFEB/3$^{KO}$ and Fam134bΔLIR cells (Fig EV4D and E). Similarly, TFEB-mediated *FAM134B* induction promoted ER-phagy in HeLa, U2-OS, and mouse embryonic fibroblast cells (Figs 6A–F and EV4F and G), suggesting that TFEB/TFE3-FAM134B axis represents a general mechanism for regulation of ER-phagy.

### FGF signaling induces ER-phagy in mice and fish cartilage

Subsequently, we tested the physiological relevance of ER-phagy regulation by FGF by analyzing femoral growth plates in mice lacking both FGFR3 and FGFR4 (FGFR3/4$^{KO}$). FGFR3/4$^{KO}$ mice were growth-retarded compared with control littermates and histological examination of the femoral growth plate unveiled an altered organization of hypertrophic chondrocytes in FGFR3/4$^{KO}$ mice (Fig 7A–C). Consistent with our *in vitro* observations, FGFR3/4$^{KO}$ growth plates showed reduced JNK1/2 activity, higher IRS1 levels, and increased mTORC1 signaling compared to control growth plates (Fig 7D). In addition, we observed reduced lysosome biogenesis and autophagy, as demonstrated by lower levels of LAMP1 and LC3II, and lysosomal and autophagy markers, respectively, in FGFR3/4$^{KO}$ growth plates compared with controls (Fig 7E). Most notably, FGFR3/4$^{KO}$ growth plates showed downregulation of *Fam134b-2* at both mRNA and protein levels (Fig 7F and G) and defective ER-phagy, demonstrated by CLIMP-63

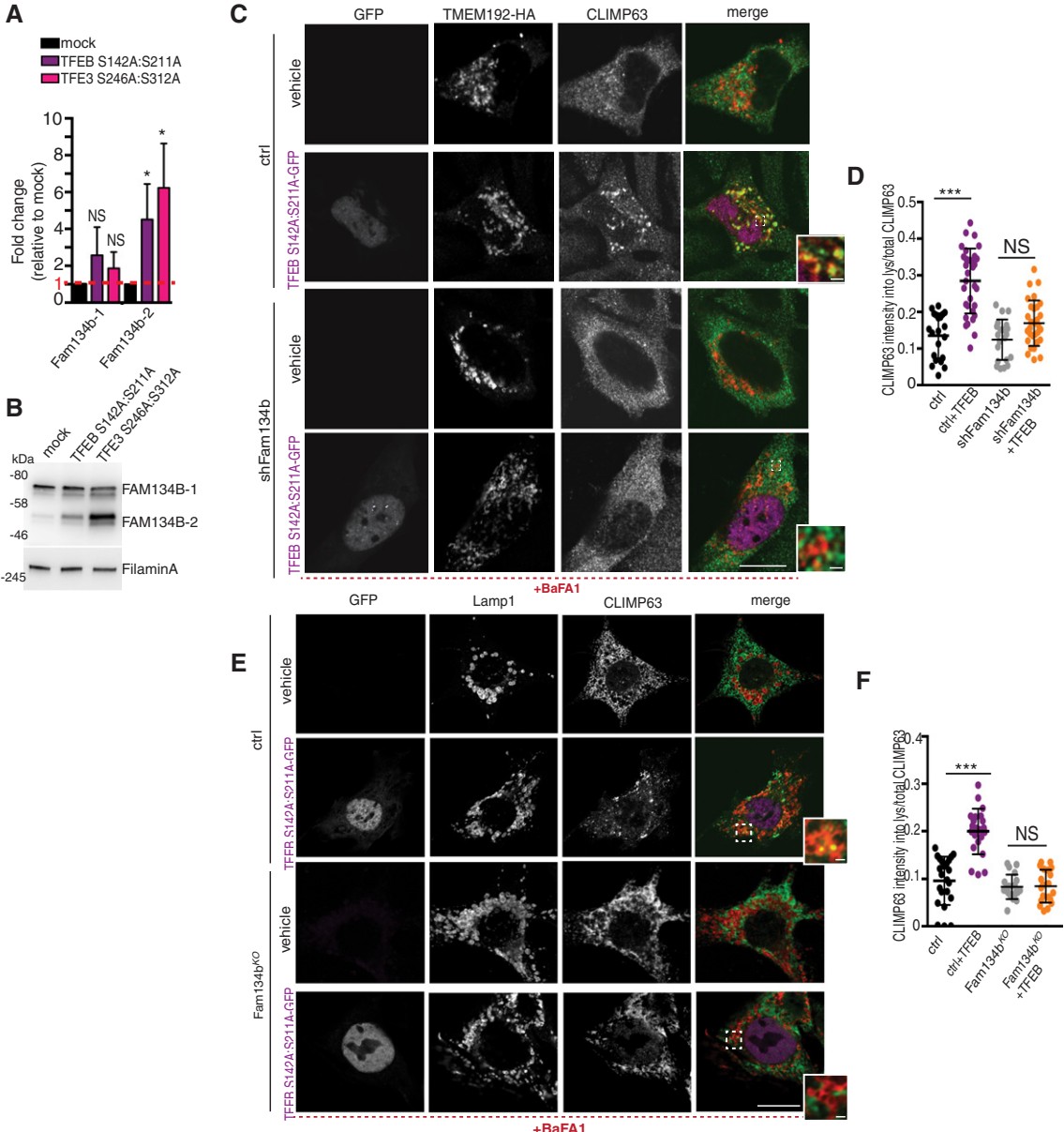

**Figure 6. TFEB overexpression induces ER-phagy in U2OS and MEFs cells via FAM134B.**

A    qRT–PCR analysis of Fam134b isoforms in mock, TFEB S142A:S211A, and TFE3 S246A:S312A overexpressing U2OS cells. Values were normalized to *Hprt* gene and expressed as fold change values relative to mock. Mean ± standard error of the mean (SEM) of N = 3 biological replicates. Student's unpaired *t*-test *P < 0.05; NS, not significant.

B    Western blot analysis of Fam134b isoforms in mock, TFEB S142A:S211A, and TFE3 S246A:S312A overexpressing U2OS cells showing induction of *FAM134B-2*, but not of *FAM134B-1* isoform by TFEB/TFE3. Filamin A was used as a loading control. Blot is representative of N = 3 independent experiments.

C, D    Co-immunofluorescence staining of ER (CLIMP-63, green) and lysosomes (TMEM192-HA, red) in control and Sh-FAM134B U2OS cells overexpressing TFEB S142A:S211A-GFP (purple). BaFA1 was used at 100 nM for 4 h. Scale bar 15 and 2 μm (higher magnification boxes) (C). In (D), quantification of CLIMP63 fluorescence in TMEM192-HA decorated lysosomes. Mean ± standard error of the mean (SEM) of N = 3 biological replicates. N = 21 (vehicle ctrl), N = 33 (TFEB S142A:S211A-GFP ctrl), N = 21 (vehicle *sh*FAM134B), N = 30 (TFEB S142A:S211A-GFP *sh*FAM134B) cells were analyzed. Student's unpaired *t*-test ***P < 0.0005; NS, not significant.

E, F    Co-immunofluorescence staining of ER (CLIMP-63, green) and lysosomes (Lamp1, red) in WT and Fam134b[KO] MEF cells overexpressing TFEB S142A:S211A-GFP (purple). BaFA1 was used at 100 nM for 4 h. Scale bar 15 and 2 μm (higher magnification boxes) (E). In (F), quantification of CLIMP63 fluorescence in Lamp1 decorated lysosomes. Mean ± standard error of the mean (SEM) of N = 3 biological replicates. N = 20 (vehicle ctrl), N = 22 (TFEB S142A:S211A-GFP ctrl), N = 21 (vehicle Fam134b[KO]), N = 20 (TFEB S142A:S211A-GFP Fam134b[KO]) cells were analyzed. Student's unpaired *t*-test ***P < 0.0005; NS, not significant.

Source data are available online for this figure.

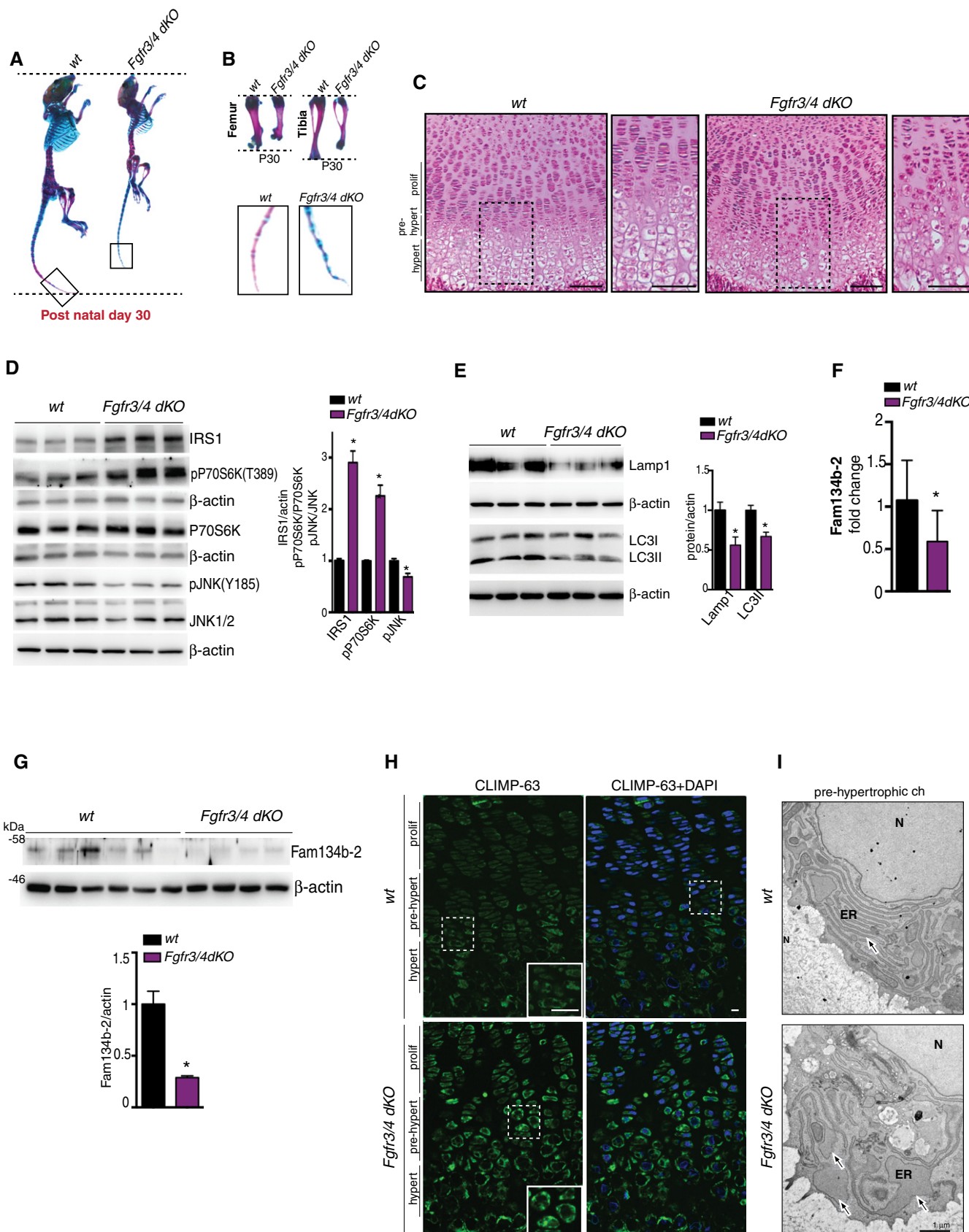

Figure 7.

◀

**Figure 7.  FGF signaling controls ER-phagy *in vivo*.**

A, B   Representative images of Alcian Blue (cartilage) and Alizarin Red (bone) skeletal staining showing growth retardation in *Fgfr3/4 dKO* mice compared to age/sex wild-type littermate at post-natal day 30. (B) Femur, tibia, and tail details.

C   Hematoxylin/eosin staining of femoral growth plate sections from wild-type and *Fgfr3/4 dKO* mice. Higher magnification insets showed a disorganized hypertrophic chondrocyte layer, in *Fgfr3/4 dKO* mice. Scale bar 60 μm.

D   Western blot analysis of IRS1, p-P70S6K (T389), P70S6K, p-JNK (Y185), JNK1/2 proteins in growth plate lysates of mice with indicated genotypes. *N* = 3 mice/genotype. β-actin was used as a loading control. Bar graph shows quantification. Mean ± standard error of the mean (SEM). Student's unpaired *t*-test *$P < 0.05$.

E   Western blot analysis of LC3 and Lamp1 proteins from growth plate lysates of mice with indicated genotypes. β-actin was used as a loading control. *N* = 3 mice/genotype were analyzed. Bar graph shows quantification. Mean ± standard error of the mean (SEM). Student's unpaired *t*-test *$P < 0.05$.

F   qRT–PCR analysis of *Fam134b-2* expression from growth plate of mice with indicated genotypes. *N* = 8 (*wt* mice) and *N* = 9 (*Fgfr3/4 dKO* mice) were analyzed. Values were normalized to *Hprt* gene and expressed as fold change relative to control. Mean ± standard error of the mean (SEM). Student's unpaired *t*-test *$P < 0.05$.

G   Western blot analysis of Fam134b-2 protein from growth plate lysates of mice with indicated genotypes. β-actin was used as a loading control. *N* = 6 (*wt* mice) and *N* = 4 (*Fgfr3/4 dKO* mice) were analyzed. Bar graph showed quantification of Fam134b-2 normalized to β-actin. Mean ± standard error of the mean (SEM). Student's unpaired *t*-test *$P < 0.05$.

H   Representative immunofluorescence staining of CLIMP-63 of femur growth plate sections from mice with indicated genotypes. Scale bar 10 μm. Insets showed increased CLIMP-63 staining in *Fgfr3/4 dKO* mice. Scale bar 5 μm. Hypert = hypertrophic chondrocytes; pre-hypert = pre-hypertrophic chondrocytes; prolif = proliferating chondrocytes.

I   Representative TEM images of growth plate chondrocytes from mice with indicated genotypes. Arrows indicated the ER. *N* = nucleus. Scale bar 1 μm.

Source data are available online for this figure.

accumulation and ER cisterna enlargement (Fig 7H and I). Collectively, these observations strongly suggest that FGF signaling is a physiological regulator of Fam134b-mediated ER-phagy in chondrocytes during endochondral ossification.

Next, we explored the consequences of ER-phagy inhibition in chondrocytes, both *in vitro and in vivo*. MS-based proteomics analysis demonstrated that the downregulation of ER proteins induced by FGF18 was significantly inhibited in Fam134b ΔLIR cells, further demonstrating the requirement of FAM134B in FGF-mediated ER-phagy (Fig 8A; Tables EV7 and EV8). Notably, the downregulation of ribosomal proteins upon FGF was also inhibited in Fam134b ΔLIR cells, suggesting that ribosome turnover is influenced by FAM134B-mediated ER-phagy (Fig 8A). Secretome analysis using tandem mass tag proteomics demonstrated that FGF-regulated protein secretion was impaired in Fam134b ΔLIR cells (Fig 8B; Table EV9). In particular, we observed that the secretion of angiogenic (VEGFs, CTGF) and matrix remodeling (Mmp13) factors, which are stimulated by FGF (Dailey *et al*, 2003), was impaired in Fam134b ΔLIR cells (Fig 8B; Table EV9). These data strongly suggest that FAM134B-mediated ER-phagy might play an important role in chondrocytes during skeletal development. To test this hypothesis, we knocked down Fam134b in medaka fish (Oryzias latipes, ol) with a specific morpholino (MO) directed against the second splice donor site (Fig EV5A and B). Consistent with our *in vitro* data, we observed increased levels of *Fam134b* and *Lamp1* mRNAs as well as an induction of ER-phagy upon FGF18 administration in wild-type medaka fish (Fig 8C–E). FGF18 failed to induce ER-phagy in *Fam134b^mo* medaka (Fig 8D and E), and TEM analysis demonstrated that *Fam134b^mo* chondrocytes had enlarged ER cisternae filled with electron-dense material. Consistent with defective ER-phagy (Forrester *et al*, 2019), biochemical analysis showed that type II procollagen accumulated in *Fam134b^mo* compared with control fish (Fig 8F and G). Phenotypically, *Fam134b^mo* fish displayed shorter body length and head size with structural abnormalities mainly restricted to bones (Fig 8H–J). Both size and mineralization of cartilage elements were severely reduced in *Fam134b^mo* fish compared with controls (Fig 8I and J). Reintroducing mRNA encoding for the human version of FAM134B tagged with HA completely rescued body length and head size, along with significant amelioration of cartilage length and ossification in morphant fish (Fig EV5C–F).

Collectively, these observations strongly suggest that the regulation of Fam134b-mediated ER-phagy by FGF is conserved across the evolution and plays important roles in chondrocytes during skeletogenesis.

## Discussion

In this study, we have described a novel pathway through which FGF signaling may regulate chondrocyte functions during endochondral ossification. By combining quantitative proteomic, phosphoproteomic, and RNA sequencing, we demonstrated that FGF stimulation promotes cellular catabolism while inhibiting anabolic pathways. This metabolic shift appears to be mediated, at least in part, by the suppression of insulin/IGF1 signaling through JNK-mediated degradation of IRS1. These data provide an additional possible mechanism for the observed inhibitory role of FGF during endochondral ossification.

We have identified the molecular players through which FGF18 triggers lysosomal degradation of ER via ER-phagy. Our data clearly show that ER-phagy is transcriptionally regulated by the MiT/TFE factors via direct induction of FAM134B, demonstrating that the nutrient-regulated signaling network downstream TFEB and TFE3 also participates to tissue development and organismal growth. Notably, FGF18 still induced autophagy flux but failed to induce ER-phagy in RCS lacking TFEB and TFE3, suggesting that, at least in this particular context, the main autophagy role of TFEB/TFE3 is to confer substrate selectivity to the autophagy process. Future studies will be needed to investigate whether additional autophagy substrates are delivered to lysosomes through similar mechanisms, or whether ER-phagy is part of the general autophagy pathway.

By increasing FAM134B levels in the ER, TFEB and TFE3 might promote FAM134B oligomerization, which in turn promote ER fragmentation and incorporation of ER-derived vesicles into nascent autophagosomes (Bhaskara *et al*, 2019; Jiang *et al*, 2020). Notably,

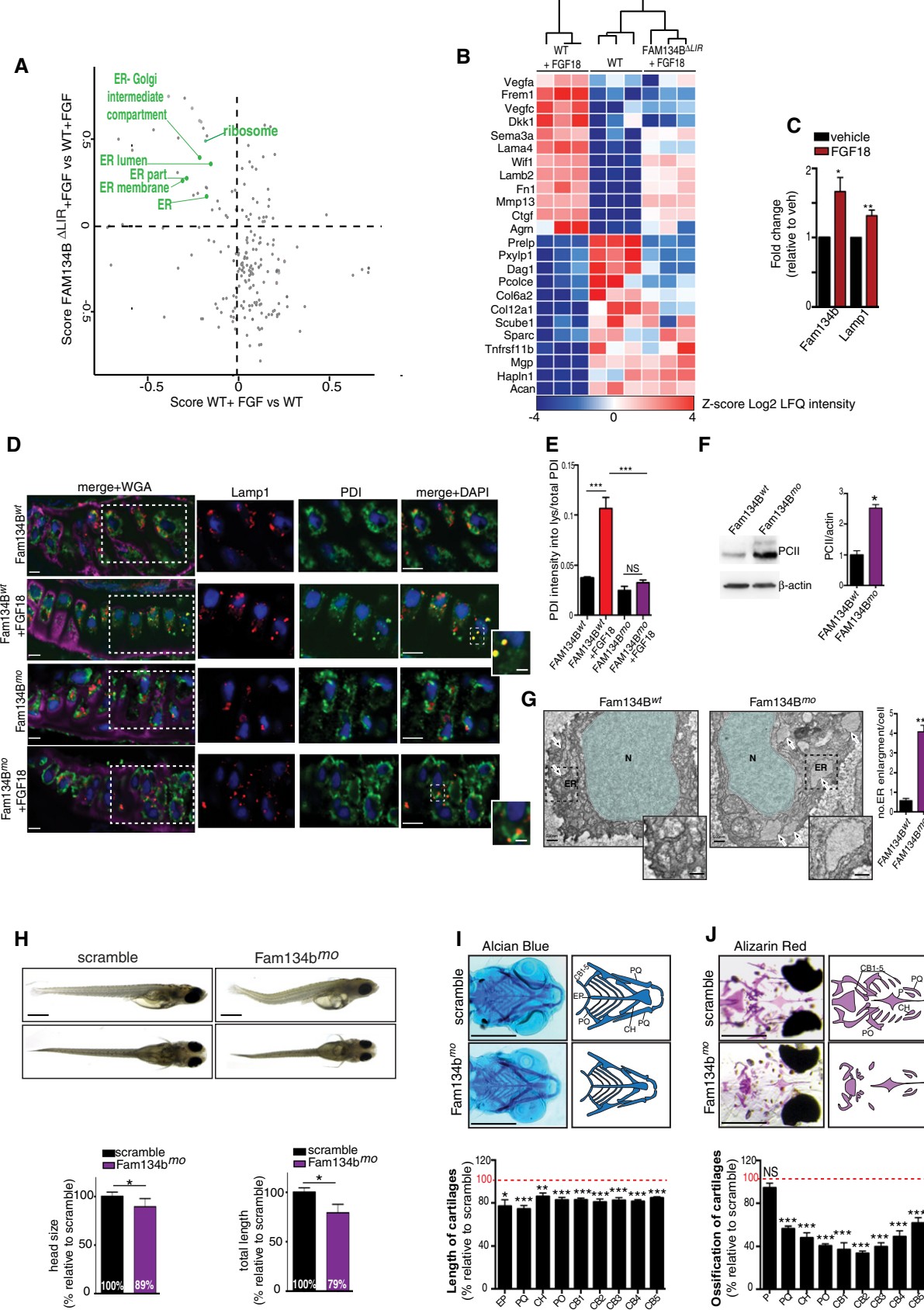

**Figure 8.**

**Figure 8.  FAM134B is required for protein secretion in RCS and skeletal growth in medaka.**

A   Scatter plot of cellular compartments and biological processes regulated by FGF18 in a Fam134b-dependent manner. Full list can be found in Tables EV7 and EV8. Score: Student's *t*-test difference between wild-type and Fam134bΔLIR cells. *N* = 3 biological replicates/treatment/genotype were analyzed. FDR < 0.05.

B   Tandem mass tag secretome analysis of chondrocytes with indicated genotypes and treatments. Fam134b-dependent secreted proteins are shown; full list can be found in Table EV9. *N* = 3 biological replicates/treatment/genotype were analyzed. FDR < 0.05.

C   qRT–PCR analysis of *Fam134b* and *Lamp1* from medaka fish embryos treated with FGF18 (50 ng/ml) for 24 h. Values were normalized to *Hprt* gene and expressed as fold change relative to untreated fish. *N* = 5 biological replicates. Mean ± standard error of the mean (SEM). Paired Student's *t*-test **$P$ < 0.005; *$P$ < 0.05.

D   Co-immunofluorescence staining of Lamp1 (lysosome) and PDI (ER) in medaka fish chondrocytes with indicated genotypes treated with FGF18 (50 ng/ml for 24 h). Insets showed details of colocalization of Lamp1 and PDI. Wheat germ agglutinin (WGA) was used to stain cartilage matrix. Representative images of *N* = 3 fish/genotype/treatment. Scale bars 10 μm. Boxed area were zoomed, and single and merged channels are shown. Scale bars 2 μm.

E   Quantification of PDI fluorescence intensity into Lamp1-positive lysosomes. Mean ± standard error of the mean (SEM) of *N* = 10 cells/experiment were analyzed. One-way analysis of variance (ANOVA) $P$ < 0.001; Sidak's multiple comparison test ***$P$ < 0.0005; NS, not significant.

F   Western blot analysis of type II procollagen from a pool of medaka fish embryos with indicated genotypes, showing procollagen accumulation in *Fam134b^{mo}* fish. β-actin was used as a loading control. Bar graph shows quantification of type II procollagen normalized to β-actin. *N* = 3 biological replicates. Mean ± standard error of the mean (SEM). Student's unpaired *t*-test *$P$ < 0.05.

G   Representative TEM images of medaka fish chondrocytes. Arrows indicated the ER. Insets show ER enlargement in *Fam134b^{mo}* chondrocyte. *N* = 3 fish/genotype were analyzed. Bar graph shows number of ER enlargement per cell in medaka fish with indicated genotypes. *N* = 29 cells (*Fam134b^{wt}*) and *N* = 37 cells (*Fam134b^{mo}*) were analyzed. Mean ± standard error of the mean (SEM). Student's unpaired *t*-test ***$P$ < 0.0005. Scale bar 500 nm. N = nucleus, ER = endoplasmic reticulum.

H   Lateral (top) and ventral (bottom) projections of medaka fish embryos with indicated genotypes. Scale bar 1 mm. Bar graphs show quantification of total length and head size of medaka fish model of *Fam134b^{mo}* expressed as % relative to the scramble. Mean ± standard error of the mean (SEM) of at least *n* = 8 fish/genotype. Student's unpaired *t*-test *$P$ < 0.05.

I, J   Alcian Blue (cartilage) (I) Alizarin Red (bone) (J) staining of scramble and *Fam134b^{mo}* medaka fish. Graph shows quantification of ethmoid plate (EP), palatoquadrate (PQ), ceratohyal (CH), paired prootics (PO), ceretobranchials 1–5 (CB1 to CB5) cartilage length (I), and bone mineralization (J) in F*am134b^{mo}* and scramble fish. Values were expressed as % relative to the scramble (100% red dotted line). Mean ± standard error of the mean (SEM) of *n* = 9 fish/genotype. Student's unpaired *t*-test *$P$ < 0.05; **$P$ < 0.005; ***$P$ < 0.0005. NS, not significant.

Source data are available online for this figure.

a transcriptional induction of *Fam134b-2* by C/EBPβ was recently shown to mediate starvation-induced ER-phagy in liver (Kohno *et al*, 2019), suggesting that multiple transcriptional circuitries might participate to the regulation of ER-phagy.

The demonstration that different extracellular cues, such as starvation and FGF18, activate ER-phagy through the same effectors suggests its role in sustaining energy metabolism in chondrocytes, which reside in a tissue environment where nutrient is scarce (Yao *et al*, 2019). In addition, ER remodeling through ER-phagy might be important in regulating protein secretion in chondrocytes, particularly during chondrocyte hypertrophic differentiation.

The discovery of the cellular mechanisms governing ER-phagy might be of therapeutic relevance. Indeed, FAM134B-mediated lysosomal degradation of ER fragments has been shown to mediate the clearance of ER-retained misfolded pathogenic proteins, such as alpha1-antitrypsin Z mutant and collagens (Fregno *et al*, 2018; Cui *et al*, 2019; Forrester *et al*, 2019). Thus, the identification of the mechanisms that control FAM134B activation might be exploited for the treatment of protein conformational diseases.

# Materials and Methods

### Cell culture, transfections, and plasmids

RCS cell line was a Swarm chondrosarcoma chondrocyte line (King & Kimura, 2003). Control and TFEB;3^{KO} HeLa cell line were previously described (Nezich *et al*, 2015). U2OS *sh*FAM134B and FAM134B KO MEF cell lines were previously described (Khaminets *et al*, 2015). RCS, MEFs, and HeLa cells were cultured in DMEM (Euroclone), supplemented with 10% fetal bovine serum (FBS from Euroclone) and 1% P/S. U2OS cells were cultured in Mc'Coys (Euroclone), supplemented with 15% fetal bovine serum (FBS from

Euroclone) and 1% P/S. In FGF18 experiments, cells were cultured in DMEM supplemented with 5% adult bovine serum (ABS from Bio-Techne) (vehicle). In starvation experiment, RCS chondrocytes were cultured in HBSS medium (from Euroclone). For transfection experiments, cells were transfected with Lipofectamine LTX and Plus reagent (Invitrogen) following reverse transfection protocol according to the manufacturer's instructions.

### Plasmids

TMEM192-HA and RAG-A/C-CA-GFP were gifts from D. Sabatini (Whitehead Institute, MIT Boston); TFEB, PRK5-TFE3, and Myc-MITF were described (An *et al*, 2019); CMV-Runx2 was a gift from G. Karsenty (Columbia University, New York); FOXO3-HA was a gift from R. Polishchuck (TIGEM, Italy); and IRS1 and Pgl3 basic luciferase plasmids were from Addgene. eGFP-mCherry-Ramp4 plasmid was a gift from E. Corn (Berkeley, California). TFEB S142A:S211-GFP TFE3 S246A:S312A-GFP, and eGFP-mCherry-LC3 plasmids were previously described (Settembre *et al*, 2011; Di Malta *et al*, 2017). FAM134B1-HA expression plasmid was described (Khanimets *et al*, 2015) and FAM13B2-HA was produced by cloning strategy.

### Generation of CRISPR clones

Disruption of genes of interest was obtained through clustered regularly interspaced short palindromic repeats (CRISPR)/CRISPR-associated protein 9 (Cas9) technology. $1 \times 10^6$ RCS chondrocytes were transfected with 5 μg of all-in-one vector containing the sgRNA of interest:
FGFR1 sgRNA sequence: GCATCGTGGAGAACGAGTATGG;
FGFR2 sgRNA sequence: TTTCGGTGTTGGTCCAGTACGG;
FGFR3 sgRNA sequence: ACGCGGGTGTCCTCAGCTACGG;
FGFR4 sgRNA sequence: TCCACGGAGAGAATCGTATCGG;

TFEB sgRNA sequence: GCTGCCATGGCGTCGCGCATCGG;
TFE3 sgRNA sequence: AGTCGTCCACCCCCTGCTC;
Fam134b sgRNA sequence: TGAGCTCTGTGGGTAAGCCAAGG;
ATG9A sgRNA sequence: CTCGTCCCGGGTCTGCGAGCGG;
Gusb sgRNA sequence: CTTCGCGGGAACTCAAGGTG.
Beclin1 sgRNA sequence: GTTTTCTGCCACCACCTTT
ATG7 sgRNA sequence: CGCTGAGGTTCACCATCCT

The all-in one vector contains the U6 promoter, a recombinant form of Cas9 protein under the control of CMV promoter, and a eGFP or mCherry reporter gene under the control of the SV40 promoter (Sigma-Aldrich). Forty-eight hours after transfection, putative clones were FACS-sorted for the eGFP or mCherry fluorescence using the BD FACSAria. Sorted cells were kept in culture until confluence and then subject to PCR analysis followed by Sanger sequencing to identify mutations. Selected clones were validated by Western blotting analysis of protein of interest.

## Chemicals

FGF ligands (PeproTech) were used at 50 ng/ml for 4 h or overnight. c-Jun N-terminal kinase (JNK) inhibitor (SP600125, Sigma-Aldrich, Milan, Italy) was used at 50 μM for 12 h. Bafilomycin A1 (Sigma-Aldrich) was used at 100 nM for 3–4 h. Torin 1 (Cell Signaling) was used at 1 μM for 2 h. Concanamycin A (Sigma-Aldrich, Milan, Italy) was used at 100 nM for 1 h. Proteasomal inhibitor (MG132, Sigma-Aldrich, Milan, Italy) was used at 10 μM for 6 h. Actinomycin (Sigma-Aldrich) was used at 1 μg/ml for 4 h. ER-Tracker BODIPY Green (BODIPY™ FL Glibenclamide, for live-cell imaging, Thermo Fisher) was used at 1 μM for 30 min at 37°C in DMEM (w/o supplements) in dark and then fixed with 4% PFA for 2 min at 37°C.

## Immunofluorescence

RCS chondrocytes were fixed for 15 min in 4% PFA in PBS and permeabilized for 20 min in blocking buffer (0.05% (w/v) saponin, 0.5% (w/v) BSA, 50 mM $NH_4Cl$, and 0.02% $NaN_3$ in PBS). Cells were incubated in humid chamber for 1 h at room temperature with primary antibodies (Lamp1 Abcam ab24170 1:200; CLIMP63 Proteintech 16686-1-AP 1:200; HA BioLegend 901501 1:500; p-IRS1 Merck 05-1087 1:100; IRS1 Cell Signaling Technology 2390 1:100; p-S6 Ribosomal protein Cell Signaling Technology 5364S 1:100; LC3 Nanotools 0231-100 1:200; Lamp1 Santa Cruz Biotechnology sc-19992 1:500), washed three times in PBS, incubated for 1 h at room temperature with the secondary (Alexa Fluor-labeled 1:400) antibodies, washed again three times in PBS, incubated for 20 min with 1 μg/ml Hoechst 33342, and finally mounted in Mowiol. All confocal experiments were acquired using slice thickness of 0.5 μm using the LSM 880 confocal microscope equipped with a 63× 1.4 numerical aperture oil objective.

### TFEB and TFE3 immunofluorescence
RCS chondrocytes were fixed for 15 min in 4% PFA in PBS and permeabilized for 30 min in 0.02% Triton X-100 in PBS. Cells were incubated in humid chamber for 1 h in blocking buffer (0.1% Triton X-100, 10% goat serum in PBS) and then with primary antibodies overnight at 4°C (TFEB MyBioSource

MBS120432 1:50; TFE3 Sigma-Aldrich HPA023881 1:200) diluted in 0.1% Triton X-100 and 5% goat serum in PBS. Alexa Fluor-conjugated secondary antibodies (1:400) were incubated for 1 h at room temperature in 0.1% Triton X-100 and 1% goat serum in PBS. Nuclei were stained with DAPI 1:1,000 in PBS for 20 min at room temperature. Cells were washed with PBS, once in Milli-Q water and mounted with Mowiol. All images were captured using LSM 880 confocal microscope equipped with a 63× 1.4 numerical aperture oil objective. All the quantifications were performed used ImageJ plugins.

## Transmission electron microscopy

For routine EM analysis, the cells were fixed with 1% glutaraldehyde (GA) prepared in 0.2 M HEPES buffer (pH 7.4) for 30 min at room temperature (RT). Mouse growth plates (P8) and medaka fish embryos (stage 40) were fixed using a mixture of 2% paraformaldehyde (PFA) and 1% GA prepared in 0.2 M HEPES buffer for 24 h at 4°C.

For immuno-EM analysis, the cells were fixed with a mixture of 4% PFA and 0.05% GA for 10 min at RT, then washed with 4% PFA once to remove the residual GA, and fixed again with 4% PFA for 30 min at RT. Next, the cells were incubated with a blocking/permeabilizing mixture (0.5% BSA, 0.1% saponin, 50 mM $NH_4Cl$) for 30 min and subsequently with the primary polyclonal antibody against GFP (Abcam, Cat No. AB 290-50) diluted 1:250 in blocking/permeabilizing solution. The following day, the cells were washed and incubated with the secondary antibody, the anti-rabbit Fab fragment coupled to 1.4 nm gold particles (Nanoprobes, Cat No 2004, anti-rabbit nanogold) diluted 1:50 in blocking/permeabilizing solution for 2 h at RT.

All specimens were then post-fixed as described in Polishchuk and Polishchuk (2019). After dehydration, the specimens were embedded in epoxy resin and polymerized at 60°C for 72 h. Thin 60-nm sections were cut on a Leica EM UC7 microtome. EM images were acquired from thin sections using a FEI Tecnai-12 electron microscope equipped with a VELETTA CCD digital camera (FEI, Eindhoven, The Netherlands). Morphometric analysis on the size of lysosomes was performed using iTEM software (Olympus SYS, Germany).

## EATR assay

$1 × 10^6$ RCS were transfected with 4 μg of eGFP-mCherry-RAMP4, and the expression of the plasmid was induced with doxycycline (Sigma-Aldrich) 4 μg/ml for 48 h. The day before, cells were treated with vehicle and FGF18 50 ng/ml for 16 h or with indicated treatments (BafA1 for 3 h 200 nM and actinomycin for 4 h at 1 μg/ml). Cells were collected in PBS, and the fluorescence was analyzed with BD FACSAria.

## GFP-mCherry-LC3 assay

$1.5 × 10^6$ RCS were transfected with 1 μg of eGFP-mCherry-LC3, and the expression of the plasmid was induced for 48 h. The day before, chondrocytes were treated with vehicle and FGF18 50 ng/ml for 16 h or with indicated treatments (BafA1 for 3 h 200 nM). Cells

were collected in PBS, and the fluorescence was analyzed with BD FACSAria.

## Western blotting

RCS chondrocytes were washed twice with PBS and then scraped in RIPA lysis buffer (20 mM Tris [pH 8.0], 150 mM NaCl, 0.1% SDS, 1% NP-40, 0.5% sodium deoxycholate) supplemented with PhosSTOP and EDTA-free protease inhibitor tablets 1× final concentration (Roche, Indianapolis, IN, USA). Cell lysates were incubated on ice for 20 min; then, the soluble fraction was isolated by centrifugation at 18,000 $g$ for 20 min at 4°C. Total protein concentration in cellular extracts was measured using the colorimetric BCA protein assay kit (Pierce Chemical Co, Boston, MA, USA). Protein extracts, separated by SDS–PAGE and transferred onto PVDF, were probed with primary antibodies overnight against IRS1 (Cell Signaling Technology 2390 1:1,000), phospho-P70S6K (Cell Signaling Technology 9234S 1:1,000), P70S6K (Cell Signaling Technology 9202S 1:1,000), phospho-AKT (Cell Signaling Technology 4056 (T308) - 4060 (S473) 1:1,000), AKT (Cell Signaling Technology 9272 1:1,000), phospho-c-JUN (Cell Signaling Technology 2361S 1:1,000), c-JUN (Cell Signaling Technology 9165 1:1,000), B-actin (Novus Biologicals NB600-501 1:5,000), Fam134b (Sigma-Aldrich HPA012077 1:1,000), FGFR1 (Cell Signaling Technology 9740S 1:1,000), FGFR3 (Cell Signaling Technology 4574S 1:1,000), FGFR4 (Cell Signaling Technology 8562S 1:1,000), FGFR2 (Cell Signaling Technology 11835S 1:1,000), TFEB (Bethyl Laboratories A303-673A 1:1,000), TFE3 (Sigma-Aldrich HPA023881 1:1,000), and p62/SQSTM1 (Abnova H00008878 1:1000). Transfected human TFEB was detected with human-specific TFEB antibody (Cell Signaling Technology BL12896_15 1:1,000). β-tubulin (Sigma T8660 1:10,000), phospho-S6 ribosomal protein (Cell Signaling Technology 5364S 1:1,000), S6 ribosomal protein (Cell Signaling Technology 2217S 1:1,000) phospho-4EBP1 (Cell Signaling Technology 9451P 1:1,000), 4EBP1 (Cell Signaling Technology 9644S 1:1,000), phospho-TFEB S142 (ABE1971 EMD Millipore 1:10,000), ATG9A (Acris Antibodies AP26284PU-N 1:1,000), type II procollagen (Hybridoma Bank II6B3 1:1,000), Lamp1 (Abcam ab24170 1:1,000), LC3 (Novus Biologicals NB100-2220 1:1,000), and Filamin A (Cell Signaling Technology 4762 1:1,000). Proteins of interest were detected with HRP-conjugated goat anti-mouse or anti-rabbit IgG antibody (1: 2,000, Vector Laboratories) and visualized with the ECL Star Enhanced Chemiluminescent Substrate (Euroclone) according to the manufacturer's protocol. The Western blotting images were acquired using the ChemiDoc-lt imaging system (UVP).

## Lysosomal enzymatic activity

Activity of lysosomal enzyme of interest (β-glucuronidase and β-hexosaminidase) was measured as previously described (Bartolomeo *et al*, 2017). Briefly, cells were lysed in extraction buffer (50 mM NaHPO$_4$ pH 7.0, 10 mM 2-mercaptoethanol, 10 mM Na$_2$ EDTA, 0.1% sodium lauryl sarcosine, 0.1% Triton X-100) and protein concentration was measured using the colorimetric BCA protein assay kit (Pierce Chemical). 200 μg of proteins was incubated with 200 μl of fluorogenic substrate (4-methylumbelliferyl-β-D-glucuronide 2 mM Sigma-Aldrich

for β-glucuronidase; 4-methylumbelliferyl-N-acetyl-β-D-glucosaminide 6 mM Sigma-Aldrich for β-hexosaminidase) for 1 h at 37°C. The reaction was stopped by adding 200 μl of the carbonate stop buffer (0.5 M NaHCO$_3$/0.5 M Na$_2$CO$_3$ pH 7.0), and the fluorescence was measured in a fluorimeter (GloMax-Multi Detection System, Promega) using 365 nm excitation and 460 emission.

## LysoTracker and DQ-BSA experiments

LysoTracker DND99 (L7528 Thermo Fisher) was incubated at 50 nM in dark for 40 min at 37°C. Cells were washed three times with PBS 1× and collected; the fluorescence was analyzed by FACS Accuri C6; and 10,000 events were collected. DQ Green BSA (D12050 Thermo Fisher) was incubated at 10 μg/ml in dark for 15 min at 37°C. Cells were washed three times with PBS 1× and collected; the fluorescence was analyzed by FACS Accuri C6; and 10,000 events were collected.

## qRT–PCR

RCS and U2OS cells were harvested for RNA extraction using RNeasy Mini Kit (Cat No./ID: 74106 (250), Qiagen) according to the manufacturer's protocol. 1 μg of total RNA was used for reverse transcription using QuantiTect Reverse Transcription Kit (Qiagen) according to the manufacturer's instructions. qRT–PCR was performed in triplicate using LightCycler 480 SYBER Green I Master (Roche) and analyzed by LightCycler 480 (Roche). The Ct values were normalized to *Cyclophillin* or *Hprt* gene, and the expression of each gene was represented as $2^{(-ddCt)}$ relative to control.

Primers used were as follows:

Rat

CtsA: Fw5′-CTTGGCTGTACTGGTCATGC-3′;
Rev5′-GGCAAAGTAGACCAGGGAGT-3′
CtsD: Fw5′-GTGGCTTCATGGGGATGGAC-3′;
Rev5′-GGAGCAAGTTGAGTGTGGCA-3′
Lamp1: Fw5′-AACCCCAGTGTGTCCAAGTA-3′;
Rev5′-GCTGACAAAGATGTGCTCCT-3′
Mcoln1: Fw5′-GTGAGCTCCAGGCCTACATTG-3′;
Rev5′-GCCACTTCCACGACGGAA-3′
Fam134a: Fw5′-CAGAACAGCAGGGTCCCATA-3′;
Rev5′-TCCACTTTAGACCCTGGCTG-3′
Fam134b: Fw5′-ACCCACAGAGCTCAAGACAA-3′;
Rev5′-CTGGTCTTTGATGGCAGCTG-3′;
Fam134c: Fw5′-CCCAGTCTTGTCCCCTGAAT-3′;
Rev5′-TTGCCTGTAGTACCACCCTG-3′
Sec62: Fw5′-TCTGGCCAGCAGAAATGAGA-3′;
Rev5′-CAGTCAGGTTTGGCAGGAAC-3′
Atl3: Fw5′-ACCCCTGCAGTTCTGTTCAC-3′;
Rev5′-CCCAGCTCAAGATACTGCCC-3′
Rtn3: Fw5′-TCTCACACACTACAGCAGCA-3′;
Rev5′- TGAGCGATGTTCACTCCTGT-3′
Ccpg1: Fw5′-TCTTGTGGCTGGACTGTCAT-3′;
Rev5′-TTTGCACTGCTTTCTCCACC-3′
Tex264: Fw5′-GTGCCAGAGGTGAAGGAGAC-3′;
Rev5′-TTGCTTGCCCCAGGAGAAAA-3′
Fam134b -2 (EX1): Fw5′-ACAGGAGGCAGTCACTTTGG-3′
Rv5′-TGCTTGCCACAACTCAGACA-3′

Fam134b -1 (EX1) Fw5′-CTACGGAGGAGCAGGAACC-3′
Rv5′-GGCCTCTTCCAGCTCAGG-3′
   Human
Fam134b -2 Fw5′-TTCATTCAAGGGAGGCAGGC-3′
Rv5′-CACCTGCTAACCACGGCTAA-3′
Fam134b -1 Fw5′-AACCTGCTGTTCTGGTTCCTT-3′
Rv5′-TCACTGAGGCTTCTCCACAAC-3′

   Mouse growth plates were lysate by tissue lyser (Qiagen) in 1 ml of TRIzol (Invitrogen) buffer for RNA extraction. RNA was isolated by chloroform phase separation and precipitated using isopropyl alcohol. RNA was eluted in RNase-free water. 1 µg of total RNA was used for reverse transcription using QuantiTect Reverse Transcription Kit (Qiagen) according to the manufacturer's instructions. qRT–PCR was performed in triplicate using LightCycler 480 SYBER Green I Master (Roche) and analyzed by LightCycler 480 (Roche). The Ct values were normalized to *Hprt* gene, and the expression of each gene was represented as $2^{(-ddCt)}$ relative to control.

   Primers used were as follows:
   Mouse
Fam134b -2 Fw5′-CATAATAGTCCACTCCTCGGCTTC-3′
Rv5′- CTCAGTCTGGCTCTTTCATCTG-3′

   Medaka fish were pooled for RNA extraction using RNeasy Mini Kit (Cat No./ID: 74106 (250), Qiagen) according to the manufacturer's protocol. 1 µg of total RNA was used for reverse transcription using QuantiTect Reverse Transcription Kit (Qiagen) according to the manufacturer's instructions. qRT–PCR was performed in triplicate using LightCycler 480 SYBER Green I Master (Roche) and analyzed by LightCycler 480 (Roche). The Ct values were normalized to *Hprt* gene, and the expression of each gene was represented as $2^{(-ddCt)}$ relative to control.

   Primers used in medaka were as follows:
Fam134b Fw5′-TCACTGCTGGAAGAAACCTG-3′
Fam134b Rev5′-ATCATGAGACGAAACCAGGG-3′
Lamp1 Fw5′-AAGTTTGGACCTGGCCACTA-3′
Lamp1 Rev5′-GTAGGTGGTGTTGGTCGGT-3′

## QuantSeq 3′ mRNA sequencing library preparation

RCS chondrocytes wild type and CRISPR-KO for the FGF receptors FGFR3 and FGFR4 (3 biological replicates/condition) were overnight cultured in 5% ABS with or without FGF18. Total RNA was extracted according to the manufacturer's instructions (RNeasy Mini Kit, Cat No./ID: 74106 (250), Qiagen); RNA extracted from both cell lines untreated was used as control. RNA extracted was quantified and mixed at 50 ng/5 µl. Total RNA (100 ng) from each sample was prepared using QuantSeq 3′ mRNA-Seq Library prep kit (Lexogen, Vienna, Austria) according to the manufacturer's instructions. The amplified fragments of cDNA (300 bp long) were sequenced in single-end mode using the NextSeq 500 (Illumina) with a read length of 75 bp.

## QuantSeq 3′ mRNA sequencing data processing and analysis

Sequence reads were trimmed using Trim Galore software () to remove adapter sequences and low-quality end bases and then aligned on rn6 reference sequence using STAR (Dobin *et al*, 2013). The expression levels of genes were determined with htseq-count (Anders *et al*, 2015) using the Gencode v19 gene model (Harrow

*et al*, 2012). The data have been deposited in NCBIs Gene Expression Omnibus (GEO) and are accessible through GEO Series accession number GSE120516. Differential expression analysis was performed using edgeR (Robinson *et al*, 2010), a statistical package based on generalized linear models, suitable for multifactorial experiments. The threshold for statistical significance chosen was false discovery rate (FDR) < 0.05: In detail, 2,225 genes were differentially expressed (1,164 genes induced and 1,061 inhibited) in the RCS-FGF18 dataset, while no one was found in the KO-RCS-FGF18 dataset (GSE120516). Gene Ontology Enrichment Analysis (GOEA) was then performed on these two lists by using the DAVID online tool (DAVID Bioinformatics Resources 6.8) restricting the output to biological process terms (BP_FAT), cellular compartment terms (CC_FAT), and molecular function terms (MF_FAT). The "Kyoto Encyclopedia of Genes and Genomes" (KEGG Pathway) analysis was also performed. The threshold for statistical significance of GOEA was FDR < 0.1 and enrichment score ≥ 1.5, while for the KEGG Pathway analysis was FDR < 0.1. The Supplementary Tables (RESULTS_GO_KEGG_DWall_RCSFGF18_ RCS and RESULTS_GO_KEGG_UPall_RCSFGF18_RCS) summarize the results of the GOEA and KEGG. Data visualization: Heatmap was generated using custom annotation scripts.

### Accession codes

The data discussed in this publication have been deposited in GEO and are accessible through GEO Series accession number GSE120516. The Series has been named: transcriptome profile of RCS chondrocyte wt cell line and RCS chondrocyte obtained from a clone ko for the FGF receptors FGFR3 and FGFR4.

### MS-proteomics and phospho-proteomics

Cells were lysed in SDC lysis buffer containing 4% (w/v) SDC, 100 mM Tris–HCl (pH 8.5). Proteome preparation was done using the in StageTip (iST) method (Kulak *et al*, 2014). Phosphopeptides were enriched using the EasyPhos workflow (Humphrey *et al*, 2015). Peptides were desalted on STAGE tips and separated on a reverse-phase column (50 cm, packed in-house with 1.9-µm C18-Reprosil-AQ Pur reversed-phase beads) (Dr Maisch GmbH) over 120 min single-run gradients at a flow rate of 350 nl on an EASY-nLC 1200 system (Thermo Fisher Scientific) and analyzed by electrospray tandem mass spectrometry on a QExactive HFX (Thermo Fisher Scientific) using HCD-based fragmentation. MS data were acquired using a data-dependent top-15 method with maximum injection time of 20 ms, a scan range of 300–1650Th, and an AGC target of 3e6. Survey scans were acquired at a resolution of 60,000. Resolution for HCD spectra was set to 15,000 with maximum ion injection time of 28 ms and an underfill ratio of either 30%. Dynamic exclusion was set to 20s. Phosphopeptides were eluted with a 140 min gradient. The maximum injection time was 20 ms, at a scan range of 300–1650Th and an AGC target of 3e6. Sequencing was performed via higher energy collisional dissociation fragmentation with a target value of 1e5 and a window of 1.6Th. Survey scans were acquired at a resolution of 60,000. Resolution for HCD spectra was set to 15,000 with a maximum ion injection time of 120 ms and an underfill ratio of 40%. Dynamic exclusion was set to 40s, and apex trigger (4–7s) was enabled.

## MS data processing and analysis

Raw mass spectrometry data were processed with MaxQuant version 1.5.5.2 using default settings (FDR 0.01, oxidized methionine (M) and acetylation (protein N-term) as variable modifications, and carbamidomethyl (C) as fixed modification. Label-free quantitation (LFQ) and "Match between runs" were enabled. For phosphoproteomic analysis, phospho-STY was used as a variable modification. Bioinformatics analysis was performed with Perseus 1.5.4.2. Annotations were extracted from UniProtKB, Gene Ontology (GO), and the Kyoto Encyclopedia of Genes and Genomes (KEGG).

The mass spectrometry proteomics data have been deposited to the ProteomeXchange Consortium via the PRIDE (Perez-Riverol *et al*, 2019) partner repository with the dataset identifier PXD015326 (RCS-FGF18/FAM134BKO-FGF18 dataset) and PXD015331 (RCS-FGF18/FGFR3-4KO-FGF18 datasets).

## Tandem mass tag secretome

Protein enrichment from cell culture supernatant was done as described before (Bonn & Otto, 2018) with minor modifications. In brief, proteins from 1 ml cell culture supernatant were bound to 20 μl primed StrataClean bead slurry by overnight incubation. Beads were washed with PBS and the proteins denatured by boiling in 2% sodium deoxycholate, 10 mM TCEP, 40 mM, chloroacetamide, and 50 mM Tris, pH 8. Samples were diluted to 1% sodium deoxycholate and digested for 6 h at 37°C with 0.5 μg trypsin; after 3 h, another 0.5 μg trypsin was added.

Tryptic peptides were enriched and desalted with SDB-RPS stage tips as described in the IST protocol (Kulak *et al*, 2014).

Peptides were separated by an 63-min non-linear gradient from 5 to 48% acetonitrile on a 20 cm self-packed C18 column with a EASY-nLC 1200 and online injected into a QExactive HF (Thermo Fisher). The mass spectrometer was operated in data-dependent mode; after a survey scan with a resolution of 60,000, the 15 most abundant ions were subjected to HCD fragmentation and analyzed with a resolution of 15,000.

## Tandem mass tag secretome analysis

Data analysis was done with MaxQuant 1.6.1 with standard parameters and activated LFQ quantification. Differentially abundant proteins were detected by 5% FDR-corrected *t*-tests performed with Perseus 1.6.2.2.

The mass spectrometry proteomics data have been deposited to the ProteomeXchange Consortium via the PRIDE (Perez-Riverol *et al*, 2019) partner repository with the dataset identifier PXD015130.

## Chromatin immunoprecipitation (ChIP)

$12 \times 10^6$ cells were transfected with pCMV-3× FLAG TFEB-WT for 48 h and treated with FGF18 overnight. Crosslinking was performed for 15 min at RT with 1% formaldehyde in PBS 1x. Formaldehyde quenching was for 5 min with 0.25 M glycine in PBS; cells were scraped on ice with cold PBS and centrifuged at 1000 g 5 min at 4°C; finally, cell lysis was performed on ice for 20 min in ChIP-Lysis buffer (50 mM Tris–HCl, pH 8, 100 mM NaCl, 1% Triton X-100, 1% Tween-20).

Chromatin shearing was performed by MNAse digestion (2 U, Sigma-Aldrich) at 37°C for 12 min. Enzymatic reaction was stopped with stop mix buffer (1% sodium dodecyl sulfate (SDS) and 2 mM ethylenediaminetetraacetic acid (EDTA)). Unbound SDS was precipitated with SDS-OUT buffer (Pierce, Rockford, IL, USA). Lysates were diluted 1:1 with ChIP buffer (50 mM Tris–HCl, pH 8, 100 mM NaCl, 0.25% Triton X-100, 2 mM EDTA) and precleared with High Capacity NeutrAvidin Agarose Resin (Pierce). Protein–DNA complexes were immunoprecipitated with 5ug of ANTI-FLAG BioM2 (Sigma-Aldrich antibody) overnight. Samples were washed three times with ChIP wash buffer (50 mM Tris–HCl, pH 8, 150 mM NaCl, 0.5% Triton X-100, 1 mM EDTA) and twice with ChIP final wash buffer (50 mM Tris–HCl, pH 8, 200 mM NaCl, 1 mM EDTA). DNA was eluted by the addition of 8 mM biotin and 1% SDS in ChIP final wash buffer. Upon crosslink reversal, DNA was precipitated overnight with 200 mM NaCl at 65°C, and purification was performed by ChIP DNA Clean &Concentrator (Zymo Research). 1 μl of DNA/qPCR was used. Primer sequences used for qPCR were as follows:

MCOLN1: FW5′-CGTCAAGCTTGTCACGTGTTC-3′
RV5′-GCGCACCGCGGTCACTG-3′
Fam134b: FW5′-CTTTTGGGTAGAGAAGTGCGTG-3′
RV5′-CACAAGCCACAGAACCCATAAT-3′

## Luciferase assay

The promoter region of rat Fam134b from 78211858 to 78211187 (CLEAR 108404, −2708 from TSS position in intron 3) was amplified by PCR from RCS chondrocyte genome and cloned into pGL3-basic luciferase reporter plasmid (Addgene). $10 \times 10^3$ cells/treatment were co-transfected with luciferase plasmid and together with 0.2, 0.5, and 1 μg of pLX304-TFEB plasmid. Luciferase assay was performed 48 h after transfection using Dual Luciferase Reporter Assay System (Promega) and normalized for transfection efficiency co-transfecting Renilla luciferase. CLEAR sequence was mutagenized using the QuickChange II mutagenesis kit (Agilent), according to the manufacturer's protocol.

## Medaka stocks

The Cab strain of wild-type medaka fish (*Oryzias latipes*) was maintained following standard conditions (i.e., 12-h/12-h dark/light conditions at 27°C). Embryos were staged according to Iwamatsu (2004). All studies on fish were conducted in strict accordance with the Institutional Guidelines for animal research and approved by the Italian Ministry of Health; Department of Public Health, Animal Health, Nutrition, and Food Safety in accordance with the law on animal experimentation (D.Lgs. 26/2014). Furthermore, all animal treatments were reviewed and approved in advance by the Ethics Committee at the TIGEM Institute (Pozzuoli, NA), Italy.

## Fam134b morpholino and mRNA injections

The available medaka *olFam134b* (DK136186) genomic sequences were retrieved from public databases (http://genome.ucsc.edu/) from human Fam134b (NM_001034850) transcript. A morpholino (Mo; Gene Tools LLC, Oregon, USA) was designed against the splicing acceptor of exon 2 (MO-Fam134b: 5′-GTCGATGATCTCCCAACT-GAAGACA-3′) of the medaka orthologous of the Fam134b gene. The

specificity and inhibitory efficiencies of morpholino were determined as previously described (Eisen & Smith, 2008). MO-Fam134b was injected at 0.015 mM concentration into one blastomere at the one-/two-cell stage. Off-target effects of the morpholino injections were excluded by repeated experiments with control morpholino or by co-injection with a p53 morpholino as previously described (Eisen & Smith, 2008; Conte *et al*, 2010). cDNA sequences corresponding to human *Fam134b* mRNA were cloned into the pcDNA 3.1(+) vector, and the corresponding synthetic mRNA was transcribed using the T7 mMessage mMachine kit (Ambion, Inc., Austin, TX, USA) according to the manufacturer's instructions. Rescue experiments were performed by co-injecting *Fam134b* mRNA (100 ng/μl) with the *olFam134b* morpholino into one blastomere of the embryos at the one-/two-cell stage.

### Cartilage and bone staining

Staining for cartilage (Alcian Blue) and bone (Alizarin Red) in fixed medaka embryos was performed according to standard medaka skeleton phenotyping protocols (https://shigen.nig.ac.jp/medaka/medakabook/index.php). Pictures were taken using the DM6000 microscopy (Leica Microsystems, Wetzlar, Germany). Measurement of both cartilage and bone length was performed using ImageJ.

Staining for cartilage (Alcian Blue) and bone (Alizarin Red) in fixed mice was performed according to the standard protocol (http://empress.har.mrc.ac.uk/browser/).

### Tissue histology and immunofluorescence

#### Mice

For histology, femur was fixed overnight in 4% (wt/vol) paraformaldehyde (PFA) and then demineralized in 10% EDTA (pH 7.4) for 48 h. Specimens were embedded in paraffin and sectioned at 7 μm, and stained with hematoxylin and eosin. For immunofluorescence, femurs were fixed overnight in 4% (wt/vol) paraformaldehyde (PFA), demineralized in 10% EDTA (pH 7.4) for 48 h, and then cryo-protected in sucrose solutions (10% sucrose for 2 h, 20% sucrose overnight, 30% sucrose over-day. All wt/vol). Specimens were embedded in OCT, and cryosections were cut at 10 μm. Sections were incubated with blocking buffer (0.3% Triton X-100, 3% BSA, 5% FBS in PBS) for 3 h at room temperature, then incubated with primary antibodies (CLIMP63 Proteintech 16686-1-AP 1:500) overnight at 4C, and finally stained with secondary antibody (Alexa Fluor-labeled 1:500) for 3 h at room temperature in blocking buffer. Images were captured using LSM 880 confocal microscope equipped with a 63× 1.4 numerical aperture oil objective.

#### Medaka fish

Embryos were subjected to anesthesia before fixation at stage 40 by 2 h of incubation in 4% paraformaldehyde, 2× phosphate-buffered saline (PBS), and 0.1% Tween-20 at room temperature (RT). Samples were rinsed three times with PTW 1× (PBS 1X, 0.1% Tween-20, pH 7.3) and then incubated overnight (ON) in 15% sucrose/PTW 1× at 4°C, and then again incubated overnight in 30% sucrose/PTW 1× at 4°C. Cryosections of the larvae were processed for immunostaining. Sections were rehydrated in PTW 1× and permeabilized in citrate buffer and boiled for 2 min. Sections were rehydrated in water and washed twice in PBS 1× for 10 min. Blocking solution containing 10% FBS/PTW 1× was applied for 1 h at RT. Primary antibodies (PDI, Enzo Life Sciences, ADI-SPA-891-F, 1:100 and Lamp1 Abcam ab24170, 1:100) were diluted in 5% FBS/PTW 1X, and sections were incubated ON at 4°C. Thus, samples were washed three times with PTW 1× and incubated with secondary antibody Alexa 647 anti-rabbit IgG, Alexa 568 anti-mouse IgG (1:200; Thermo Fisher), and WGA (1:500) in 5% FBS/PTW 1× for 2 h at RT. Nuclei were stained with DAPI 1:700 in PBS for 10 min at RT. Finally, sections were washed three times with PTW 1X and mounted with PBS/glycerol solution.

## Data availability

The proteomic data that support the findings of this study have been deposited in PRIDE under accession codes PXD015326 (http://www.ebi.ac.uk/pride/archive/projects/PXD015326), PXD015331 (http://www.ebi.ac.uk/pride/archive/projects/PXD015331), and PXD015130 (http://www.ebi.ac.uk/pride/archive/projects/PXD015130). The QuantSeq data that support the findings of this study have been deposited in the Gene Expression Omnibus (GEO) under accession code GSE120516 (https://www.ncbi.nlm.nih.gov/geo/query/acc.cgi?acc=GSE120516). Source data and quantifications given in the main text have associated raw data. All other data supporting the findings of this study are available from the corresponding authors on reasonable request.

**Expanded View** for this article is available online.

### Acknowledgements

We thank G. Diez Roux and MA. De Matteis (TIGEM) for suggestions and critical reading of the manuscript. We thank D.M. Sabatini (Whitehead Institute, Boston, USA), G. Karsenty (Columbia University, New York, USA), R. Polishchuck (TIGEM, Pozzuoli, ITA), and E. Corn (University of Berkeley, San Francisco, USA) for sharing reagents. This work was supported by grants to CS: European Research Council (ERC) starting grant (714551) and Associazione Italiana per la Ricerca sul Cancro (AIRC) (A.I.R.C.) (IG 2015 Id 17717); NK is funded by Emmy-Noether DFG (KR5166/1-1).

### Author contributions

LC performed most of the experiments. CDL performed medaka and JNK/IRS experiments. MI and MS performed transcriptional analysis of FAM134b. MMa performed enzymatic assays of lysosomal proteins. MI, CDM, and MC performed ChIP experiments. NK, FS, PG, and MMa performed and analyzed proteome and phospho-proteome data. RDC performed Quantisec analysis. LGW helped with FACS analysis. EP performed electron microscopy analysis. CLa performed mouse genotyping and plasmid preparations. IC, ID, and FGS generated and validated medaka model. FB and ID performed secretome analysis. PG, AB, AKH, and CAH provided reagents and suggestions. CS designed the study and supervised the experiments. CS, LC, and CDL wrote the paper and prepared the figures.

### Conflict of interest

The authors declare that they have no conflict of interest.

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
