## [Review Process File · The EMBO Journal]

MiT/TFE factors control ER-phagy via transcriptional regulation of FAM134B

Laura Cinque, Chiara De Leonibus, Maria Lavazzo, Natalie Kraemer, Daniela Intartaglia, Francesco Salierno, Rossella De Cegli, Chiara Di Malta, Maria Svelto, Carmela Lanzara, Marianna Maddaluno, Luca Wanderlingh, Antje Hübner, Marcella Cesana, Florian Bonn, Elena Polishchuk, Christian Hübner, Ivan Conte, Ivan Dikic, Matthias Mann, Andrea Ballabio, Francesca Sacco, Paolo Grumati, and Carmine Settembre

DOI: [10.15252/emj.2020105696](https://doi.org/10.15252/emj.2020105696)

Review Timeline:

Submission Date:	25th May 20
Editorial Decision:	2nd Jun 20
Revision Received:	11th Jun 20
Accepted:	18th Jun 20

Editor: Elisabetta Argenzio

Transaction Report:

The initial review process for this manuscript took place with another journal. The initial reviewers' comments and authors' responses for this article have been made available. All referees have been contacted and agreed to have their comments published with this article.

Reviewers' comments

Reviewer #1 (Remarks to the Author):

This study reports that FGF18 signalling via FGFR3/4 enhances ER-phagy in chondrocytes. This paper is partly an extension of an earlier study by the same group (ref 7) reporting that FGF7 signalling induces autophagy in chondrocytes. This current study suggests two mechanisms for the effects of FGF18. The first is that this causes mTORC1 inhibition via JNK-mediated degradation of IRS1. This mechanism can explain how FGF18 induces TFEB/TFE3 activities and activates lysosomal biogenesis and autophagy. The second mechanism is that FGF18 may induce the expression of FAM134, an ER-phagy receptor by binding a TFEB/TFE3 binding site in its promoter.

Major comments:

1. This is not the first report of transcriptional regulation of ER-phagy in response to physiological signals. Kohno et al (Life Sci Alliance 2:pii: e201900340 have reported something analogous in the liver in response to starvation mediated by an N-terminally truncated isoform of FAM134B.
2. It is unclear how JNK signalling results in more rapid turnover of IRS1 – there is a gap in the mechanism. Is this direct or indirect?
3. The authors have not provided evidence that conclusively demonstrates that TFEB/TFE3 binding to the promoter of FAM134B is driving its transcription in response to FGF18 or that that this is necessary for the enhanced ER-phagy. In Figure Supp 4e, there is still an effect of FGF stimulation in TFEB/TFE3-knockout cells. In order to test the hypothesis, the authors should examine the effect of FGF18 on the “complete” FAM134B promoter either in its wild-type form or with the CLEAR site mutated. Given that FGF18 also activates the beclin 1 complex (ref 7), the authors should consider mutating the CLEAR site in the endogenous FAM134B promoter to test if this impacts ER-phagy responses to FGF18.
4. How generalizable are these effects to other cell types?

Technical comments:

1. The authors should show representative experimental data for CHIP assays.
2. Given the paper of Kohno et al referenced above, the authors should make attempts to distinguish if they are assessing FAM134B-1 or FAM134B-2, as this may have physiological consequences.

Reviewer #2 (Remarks to the Author):

Cinque et al have previously shown that FGF18 regulates autophagic flux in chondrocytes during development, in order to maintain collagen secretion. Here they show that FGF18 regulates specifically ER-phagy - via an Insulin Receptor-mTOR-TFEB pathway - and that this is also a major component of the mechanism for metabolic amino acid/mTOR regulated ER-phagy, as is commonly studied in the literature. The manuscript is novel, well written and provides major insight into upstream regulation of ER-phagy, which has been lacking in the field. Statistics seem fine and appropriate referencing is used.

However, while the signalling pathway and the connections between the components thereof seems plausible, I think it would be important to know the scope of ER-phagy function downstream of this physiologically. The experiment with the Medaka fish seems an important first step toward this but in itself is perhaps not quite sufficient. In particular, could some extra evidence perhaps be gained along the following lines prior to publication?:

- 1) Does ER-phagy affect collagen secretion via the TFEB-FAM134 axis in chondrocytes (in vitro or in vivo) and/or does it play a role in FGF18-mediated chondrocyte differentiation (in vitro or in vivo) downstream of this axis?
- 2) In the authors' previous work they generated FGF18-deficient mice with growth plate defects. Is it possible to use this or another in vivo system to observe FGF18-mediated regulation of FAM134 transcripts in vivo?

Other important technical considerations:

3) It's not yet wholly accepted that FAM134B-2, missing much of the reticulon domain, is a potent autophagy inducer (despite the Life Science Alliance Paper). In any case, the authors should rescue the ER-phagy deficiency upon FGF18 treatment in FAM134B-LIR cells, or other, by re-expressing FAM134B-2. In any event, the statement on page 6 about the function of the reticulon domain of FAM134B may need qualified in this light.

MINOR POINTS

1) Last paragraph of main text – the clearance of anti-1-trypsin by FAM134B in the referenced model is not true ER-phagy.

Reviewer #3 (Remarks to the Author):

In the manuscript by Cinque et al, the authors showed that FGF18 regulates ER-phagy in chondrogenic cell line. They further revealed the underlying pathway, which includes JNK-IRS1-Akt-mTORC1 signaling, converging on TFE3/TFEB transcription factors, which in turn regulate expression of Fam134b, a receptor for selective autophagic elimination of the endoplasmic reticulum (ER-phagy). The manuscript is well written, methodologically sounds and addresses important general question of mechanisms underlying ER-phagy regulation. However, I have several concerns specified below and without addressing them I feel that the manuscript is overstated and over-generalized.

Major points:

1. The authors clearly and convincingly demonstrated that in RCS chondrosarcoma cell line FGF18 stimulates TFEB/TFE3-mediated induction of Fam134b expression. This is a very interesting, novel and, to my knowledge, previously undescribed connection. However, generality of this observation is limited to RCS chondrosarcoma cell line and confirmation in other cell lines or in vivo would add strength and depth to this observation. I would suggest to add MEFs and Saos2 (used by the authors in their recent study EMBO J 2019, 38:e99847) for verification of this connection. This would allow to make more generalized conclusion. Alternatively, in vivo experiments can be expanded (please see my comment below to the presented in vivo data).
2. The authors showed that regulation of TFEB/TFE3 by FGF18 is mediated via JNK-dependent degradation of IRS1 leading to down-regulation of AKT-mTORC1 pathway. FGF18-dependent activation of JNK is known and particularly for RCS cells has been demonstrated by these same authors previously (Cinque L et al., Nature 2015, 528:272-276). In relation to AKT-mTORC1 pathway, previously the authors claimed that there is no regulation of Akt/mTORC1 by FGF18 in RCS cells (Suppl Fig 8, Nature 2015, 528:272-276), whereas in the current manuscript they showed down-regulation of Akt/mTORC1 activity by FGF18 (Fig. 4c,d). This clear discrepancy requires clarification.
3. The authors claimed that FGF18 specifically regulates ER-phagy in RCS cells via TFEB/TFE3-Fam134b axis. TFEB/TFE3 are master regulators of autophagy and lysosomal biogenesis in general. Additionally, in the current manuscript (Fig. 1b,c,d,h) and previously (Cinque L et al., Nature 2015, 528:272-276), the authors showed that FGF18 increases general levels of autophagy. Thus, since ER-phagy is a sub-type of the autophagy process, the observed increase in ER-phagy can reflect overall increase in the autophagy and lysosomal biogenesis upon FGF18 stimulation, not selective ER-phagy. To claim selective regulation of ER-phagy by FGF18, changes in the ratio of ER-phagy to autophagy levels should be demonstrated, e.g., EATR assay normalized per LC3-puncta or the number of lysosomes.
4. To explore physiological role of Fam134b the authors did morpholino ablation of this gene in medaka fish and claimed its physiological role in endochondral ossification (Fig 2i,j and page 4, paragraph 1). First, there is no endochondral ossification in cranial skeleton of medaka. Bones are formed intra-membranously around cartilaginous skeletal elements, like mandibular bone around Meckel's cartilage and no claims about chondrocyte differentiation can be made. Second, the data are very preliminary, levels of Fam134b, lysosomes and CLIMP-63 puncta have to be demonstrated in medaka's chondrocytes with/without Fam134b ablation as well as morphological and transcriptional changes in chondrocytes to claim physiological role of Fam134b in chondrogenesis.
5. Another important point is related to the physiological role of Fam134b – neither mice deficient in Fam134b nor patients with mutated Fam134b were reported signs of growth retardation. This should be discussed.
6. In light of the comment above the title is not appropriate. There is no data supporting chondrocyte differentiation. Mentioning chondrocytes in the title is also overstating since only chondrosarcoma cell line was used.

Minor comments:

- 1) Throughout the manuscript the authors made an impression that FGF18 is signaling exclusively via FGFR3 and FGFR4, whereas FGFR1 and FGFR2 are not mediating this signaling. This can be concluded only if the expression of FGFR1 and/or FGFR2 is demonstrated in this particular cell line. The alternative is that RCS cells do not express FGFR1 and FGFR2.
- 2) In Fig. 3C only TFE3 is significantly upregulated whereas in the text the authors say that TFE3, TFEB and MITF are significantly upregulated. This should be corrected.
- 3) Numerous Western blots lack quantification and it is unclear how reliable are those observations.
- 4) FGF18 downregulated ribosomal categories by mass-spec, but upregulated their gene expression (Fig. 1a and Fig. S3c). It would be good to discuss if this is a sign of selective elimination of ribosomes by ER-phagy or methodological discrepancy.

Authors' Response - Round 1

We thank the editors and reviewers for their constructive comments, and for the opportunity to reconsider our work. The revised version will contains several new data (please see the rebuttal figures) that reinforce our original findings, demonstrate physiological relevance of our results and broaden their significance. In particular, we demonstrated that: 1) FGF signalling stimulates ER-phagy *in vivo*, in cartilage of both mouse and medaka fish; 2) the activation of ER-phagy is required for the FGF-mediated secretion in chondrocytes; 3) mice lacking fibroblast growth factor receptors 3 and 4 (FGFR3 and FGFR4) show altered PI3K signalling, impaired FAM134B expression and, in turn, defective ER-phagy in chondrocytes. Notably, these mice have enlarged ER, impaired hypertrophic differentiation of chondrocytes and severe bone growth retardation. 4) TFEB/TFE3 activation induces FAM134B-mediated ER-phagy in multiple cell types and experimental conditions, demonstrating the broader biological context of the pathway we describe.

Thus, our work sheds light on an entirely new, evolutionarily conserved, and physiologically relevant pathway that regulates ER-phagy during organismal development and in response to metabolic demands.

In addition, we provide a point-by-point response to the reviewers concerns that addresses all issues raised during the revision.

Reviewer #1 (Remarks to the Author): this study reports that FGF18 signalling via FGFR3/4 enhances ER-phagy in chondrocytes. This paper is partly an extension of an earlier study by the same group (ref 7) reporting that FGF7 signalling induces autophagy in chondrocytes. This current study suggests two mechanisms for the effects of FGF18. The first is that this causes mTORC1 inhibition via JNK-mediated degradation of IRS1. This mechanism can explain how FGF18 induces TFEB/TFE3 activities and activates lysosomal biogenesis and autophagy. The second mechanism is that FGF18 may induce the expression of FAM134, an ER-phagy receptor by binding a TFEB/TFE3 binding site in its promoter.

Major comments:

1) This is not the first report of transcriptional regulation of ER-phagy in response to physiological signals. Kohno et al (Life Sci Alliance 2:pii: e201900340 have reported something analogous in the liver in response to starvation mediated by an N-terminally truncated isoform of FAM134B.

(Response) The work by Kohno et al. demonstrated that FAM134B-2 is regulated by C/EBPb transcription factor in liver during starvation. This important work was cited in our manuscript, and discussed in this revised version.

Our work identifies FGF signalling as a major regulator of ER-phagy in chondrocytes and defines all the molecular players through which this regulation occurs. This signalling pathway was never described before. Notably, our work also demonstrates, for the first time, that TFEB is involved in the transcriptional regulation of ER-phagy in chondrocytes, as well as in other cell types. We also provide physiological significance to our findings during bone development and growth. It is important to emphasize that FGF represents a major regulator of skeletal development and when deregulated causes at least 15 different type of genetic skeletal disorders.

2) It is unclear how JNK signalling results in more rapid turnover of IRS1 – there is a gap in the mechanism. Is this direct or indirect?

(Response) It is well established that proteasomal degradation of IRS1 can be stimulated by phosphorylation operated by different kinases, such as mTORC1, S6K, IKKb, and JNK (reviewed in Haeusler RA et al. NRCMB 2018). In particular, JNK1/2 can stimulate IRS1 degradation mainly through phosphorylation of the serine residue 307 (S307). Consistently, we found that FGF18 induces IRS1 phosphorylation at S307 and this event precedes IRS1 degradation. Both, IRS1 S307 phosphorylation and degradation can be inhibited by using a JNK1/2 inhibitor. These observations support the model by which FGF18 induces IRS1 degradation via JNK1/2 direct phosphorylation on S307 (Rebuttal fig. 1) (in the manuscript: Supplementary fig 2b).

Rebuttal figure 1

3) The authors have not provided evidence that conclusively demonstrates that TFEB/TFE3 binding to the promoter of *FAM134B* is driving its transcription in response to FGF18 or that that this is necessary for the enhanced ER-phagy. In Figure Supp 4e, there is still an effect of FGF stimulation in TFEB/TFE3-knockout cells. In order to test the hypothesis, the authors should examine the effect of FGF18 on the “complete” *FAM134B* promoter either in its wild-type form or with the CLEAR site mutated. Given that FGF18 also activates the beclin 1 complex (ref 7), the authors should consider mutating the CLEAR site in the endogenous *FAM134B* promoter to test if this impacts ER-phagy responses to FGF18.

(Response) The first version of the manuscript provided the following experimental evidences to demonstrate that TFEB/TFE3 binding to the promoter of *FAM134B* is driving its transcription in response to FGF18:

- 1) TFEB/TFE3 overexpression enhanced, while TFEB/TFE3 deletion blunted, *FAM134B* mRNA induction by FGF18 (now Fig.4d). Please note that the experiments that show *FAM134B* induction by FGF18 in control and TFEB/TFE3 dKO cells were reported in Fig. 3d, and not in Supp 4e. The bar graph clearly showed that the induction of *FAM134B* mRNA levels by FGF18 is largely mediated by TFEB/TFE3. Consistently, FGF18 failed to induce *FAM134B* protein levels in TFEB/TFE3dKO cells (now Fig.4e). The residual upregulation of *FAM134B-2* mRNA levels observed in TFEB/TFE3 KO cells treated with FGF18, might be due to the activity of other members of the Mi-TFE family (e.g. MiTF) or other transcription factors (e.g. C/EBPb). These data clearly demonstrate that TFEB and TFE3 are main transcriptional regulators of *FAM134B* induction by FGF18 in chondrocytes.
- 2) Next to test whether TFEB/TFE3 control *FAM134B* expression directly, we performed a bioinformatic analysis to identify TFEB/TFE3 binding (CLEAR) sites in the *FAM134B* locus. Different CLEAR sites were predicted to be upstream the 5' UTR of *FAM134B-2* isoform. Chromatin Immunoprecipitation (ChIP) analysis demonstrated that TFEB binds to a CLEAR site located 2.7kb upstream of the 5' UTR of *FAM134B-2* (Fig 4j).
- 3) Luciferase reporter assay using as promoter a 700bp DNA fragment containing the CLEAR site (CLEAR-wt) demonstrated that TFEB promotes, in a dose dependent manner, luciferase expression (Fig 4 k-l).

As also recognized by reviewer 3 (major point 1), these data clearly and convincingly demonstrated that in the RCS cell line, FGF18 stimulates TFEB/TFE3-mediated induction of *Fam134b* expression. To further support our findings, the revised version of the manuscript contains the following new experiments:

- 4) Mutation of the CLEAR site (CLEAR-mut) completely abrogated luciferase induction by TFEB (Rebuttal fig 2a-b) (in the manuscript: Fig 4k-m).
- 5) Luciferase reporter assay demonstrating that FGF18 boosts TFEB transactivation potential by using the CLEAR-wt, but not the CLEAR-mut promoter (Rebuttal Fig 2b-c) (in the manuscript: Fig 4k-m)

- 6) FGF18 does not significantly transactivate the CLEAR-wt luciferase promoter in TFEB/TFE3 dKO cells. (Rebuttal Fig 2c) (in the manuscript: Fig 4k-m)
- 7) As per reviewer suggestion we have examined the effect of FGF18 on the “complete” FAM134B promoter. We have cloned a DNA fragment of 3280 bp upstream the 5'UTR of Fam134b-2 either in its wild-type form or with the CLEAR site mutated. We found that FGF18 has an impaired capacity to transactivate the luciferase plasmid harbouring a mutated version of the CLEAR site (Rebuttal Fig 2d).

Rebuttal figure 2

- 8) We analysed unpublished TFEB ChIP-seq experiments performed in starved HeLa and ARPE19 cells. Both experiments demonstrated that TFEB, either endogenous (in HeLa) or overexpressed (in ARPE19), binds to several CLEAR elements in the *FAM134B* locus (Rebuttal fig 3). These observations are also consistent with the new data that show that TFEB/TFE3 enhance FAM134B-mediated ER-phagy in multiple cell lines.

Rebuttal figure 3

Unfortunately, to date we have not been able to mutate the endogenous CLEAR site in the promoter of *Fam134b*. However, we believe that, taken together, our data provide convincing evidences demonstrating that FGF18 stimulates *Fam134b* expression via TFEB/TFE3.

4) How generalizable are these effects to other cell types?

(Response) The following evidences demonstrate that the TFEB/TFE3-FAM134B axis represents a general mechanism that regulates ER-phagy:

1) An analysis of the expression levels of *FAM134B* in HeLa wild type and TFEB/TFE3 double-KO cells, revealed that the *FAM134B* mRNA induction was significantly impaired in TFEB/TFE3 dKO compared with WT cells upon starvation (Rebuttal fig. 4a). Consistently, starvation-induced ER-phagy was also impaired in TFEB/TFE3 dKO cells, as demonstrated by Climp63 lysosomal localization and EATR assay (Rebuttal fig. 4b-d) (in the manuscript: Supplementary Fig. 8f,g). These data demonstrate that TFEB/TFE3 regulate starvation induced ER-phagy in HeLa cells.

Rebuttal figure 4

2) Overexpression of a constitutively active form of TFEB (S142A:S211A mutant) strongly increased *FAM134B* mRNA expression levels and induced ER-phagy (Climp63 accumulation in lysosomes) in U2-OS cells, demonstrating that TFEB overexpression is sufficient to trigger ER-Phagy (Rebuttal Fig 5a and c-d). More importantly, TFEB overexpression in U2-OS cells in which *FAM134B* was stably downregulated (U2-OS Sh-FAM134B) failed to trigger ER-phagy (Rebuttal Fig 5b-d) (in the manuscript: Supplementary Fig. 9a-d)

Rebuttal figure 5

3) Overexpression of a constitutively active form of TFEB induced ER-phagy (Climp63 delivery in lysosomes) in WT, but not FAM134BKO mouse embryonic fibroblasts (Rebuttal Fig 6a,b) (in the manuscript: Supplementary Fig. 9e,f).

Rebuttal figure 6

4). FGF18 administration to Medaka fish induces *Fam134b* mRNA expression and ER-phagy (PDI delivery into LAMP1 positive vesicles) in chondrocytes of WT but not of moFAM134B fish (Rebuttal Fig 7a-c) (in the manuscript: Supplementary Fig. 6c-e).

Rebuttal figure 7

Taken together these data clearly demonstrated that the TFEB-FAM134B axis is a general mechanism through which ER-phagy can be regulated in response to cellular needs.

Technical comments:

1. The authors should show representative experimental data for CHIP assays.

(Response) A representative gel electrophoresis of the CHIP experiment is shown in Rebuttal fig 8.

Rebuttal figure 8

2. Given the paper of Kohno et al referenced above, the authors should make attempts to distinguish if they are assessing FAM134B-1 or FAM134B-2, as this may have physiological consequences.

(Response) We observed that both FAM134B isoforms are expressed and respond to TFEB in RCS, although isoform 2 is most effectively upregulated by starvation and FGF18 stimulation (now supplementary fig 7d). *In vivo*, in chondrocytes FAM134b-2 expression is predominantly regulated by FGF signalling, since it is significantly downregulated, both at mRNA and protein levels) in the growth plates of FGFR3/4 dKO mice compare to controls (Rebuttal fig 9a, b) (in the manuscript: Fig. 5f,g).

Rebuttal figure 9

In human cell lines, such as U2-OS we observed that TFEB and starvation predominantly regulate FAM134B-2 isoform. Chip-seq experiments shown in Rebuttal fig. 3 demonstrates that TFEB binds the promoters of both isoforms. Thus, consistent with Kohno et al, there is a cell- and tissue-specific transcriptional regulation of FAM134B isoforms.

FAM134B-2 is a truncated form that lacks a portion of the Reticulon Homology Domain (RHD) of FAM134B-1. We demonstrated that both isoforms correctly localize to the ER (Rebuttal fig 9a), induce ER fragmentation and ER-phagy, although with different efficiencies (Rebuttal fig 10b-c), consistent with previous observations (Bhaskara et al. 2019).

Rebuttal figure 10

Reviewer #2 (Remarks to the Author):

Cinque et al have previously shown that FGF18 regulates autophagic flux in chondrocytes during development, in order to maintain collagen secretion. Here they show that FGF18 regulates specifically ER-phagy - via an Insulin Receptor-mTOR-TFEB pathway - and that this is also a major component of the mechanism for metabolic amino acid/mTOR regulated ER-phagy, as is commonly studied in the literature. The manuscript is novel, well written and provides major insight into upstream regulation of ER-phagy, which has been lacking in the field. Statistics seem fine and appropriate referencing is used.

(Response) We thank the reviewer for recognizing the novelty and the quality of our work.

However, while the signalling pathway and the connections between the components thereof seems plausible, I think it would be important to know the scope of ER-phagy function downstream of this physiologically. The experiment with the Medaka fish seems an important first step toward this but in itself is perhaps not quite sufficient. In particular, could some extra evidence perhaps be gained along the following lines prior to publication?:

(Response) We performed several new experiments to address the scope of ER-phagy both *in vitro* and *in vivo* in both medaka and mice.

1) Does ER-phagy affect collagen secretion via the TFEB-FAM134 axis in chondrocytes (in vitro or in vivo) and/or does it play a role in FGF18-mediated chondrocyte differentiation (in vitro or in vivo) downstream of this axis?

(Response) To investigate whether ER-phagy regulates protein secretion in chondrocytes, we performed Tandem Mass Tag spectrometry to analyse the secretome in control and FAM134B^{ΔLifR} chondrocytes upon FGF stimulation. We found that FGF regulates, in a FAM134B-dependent manner, the secretion of collagens (Col6a2, Col12a1), angiogenic factors (VEGFa/VEGFc), matrix metalloproteases (MMP13) and chondrogenic/osteogenic factors (MGP and SEMA3A) (Rebuttal fig

11a). We have not been able to study Col2 trafficking/secretion in FGF18-treated RCS cells since we found that FGF18 strongly suppresses the transcription of this specific procollagen in these cells (Rebuttal fig 11b-c). However, *in vivo*, medaka fish lacking FAM134B (moFAM134B) showed aberrant accumulation of type II collagen and ER distension in chondrocytes compared to control fish (Rebuttal fig 11d and e), consistent with a defective secretion in absence of ER-phagy. Taken together these data demonstrate that ER-phagy induction is required for a proper FGF-regulated secretion in chondrocytes. (in the manuscript: Fig. 6b,f,g)

Rebuttal figure 11

2) In the authors' previous work they generated FGF18-deficient mice with growth plate defects. Is it possible to use this or another in vivo system to observe FGF18-mediated regulation of FAM134 transcripts in vivo?

(Response) We have addressed this request both in Medaka and in mice, Rebuttal figures 12 and 13, respectively.

We demonstrated that FGF18 administration to medaka fish increases both *Fam134b* and *Lamp1* mRNA levels (Rebuttal fig 12a) and induces ER-phagy, as demonstrated by accumulation of protein disulfide-isomerase (PDI) in LAMP1 positive vesicles. Importantly, FGF18-mediated ER-phagy is impaired in moFAM134B fish (Rebuttal fig 12b-c) (in the manuscript: Fig. 6c-e).

Rebuttal figure 12

In agreement with our *in vitro* findings, growth plate extracts from FGFR3/4-dKO mice showed reduced JNK1/2 activity, IRS1 accumulation, higher mTORC1 signalling (Rebuttal fig 13a) and lower levels of *FAM134B-2* mRNA (Rebuttal fig 13b) and protein (Rebuttal fig 13c) compared to control littermates. FGFR3/4dKO growth plate chondrocytes also showed defective lysosome biogenesis and autophagy, as assessed by LAMP1 and LC3II levels (Rebuttal fig 13d), respectively, ER cisternae distension and Climp63 accumulation, suggesting defective ER-phagy (Rebuttal fig 13e-g) (in the manuscript: Fig. 5).

Rebuttal figure 13

The FGFR3/4dKO mice showed severe bone growth retardation and a growth plate phenotype characterized by disorganized hypertrophic chondrocytes (Rebuttal fig 14a,b). To our knowledge, this is the first phenotypic characterization of growth plate phenotype in FGFR3/4 dKO. Taken together these *in vivo* observations clearly demonstrate the physiological relevance and evolutionarily conservation of the FGF-mediated regulation of ER-phagy. (in the manuscript: Fig. a-c)

Rebuttal figure 14

Other important technical considerations:

3) It's not yet wholly accepted that FAM134B-2, missing much of the reticulon domain, is a potent autophagy inducer (despite the Life Science Alliance Paper). In any case, the authors should rescue the ER-phagy deficiency upon FGF18 treatment in FAM134B-LIR cells, or other, by re-expressing FAM134B-2. In any event, the statement on page 6 about the function of the reticulon domain of FAM134B may need qualified in this light.

(Response) We performed the experiment suggested by the reviewer, and demonstrated that FAM134B-2 localizes at the ER, and induces ER-phagy in FAM134B-LIR cells, although less efficiently than the FAM134B-1 isoform (Rebuttal fig 15 and see also Rebuttal fig 10). These data are consistent with Bhaskara et al. 2019.

Rebuttal figure 15

MINOR POINTS

Last paragraph of main text – the clearance of anti-1-tryptin by FAM134B in the referenced model is not true ER-phagy.

(Response) We corrected this sentence.

Reviewer #3 (Remarks to the Author):

In the manuscript by Cinque et al, the authors showed that FGF18 regulates ER-phagy in chondrogenic cell line. They further revealed the underlying pathway, which includes JNK-IRS1-Akt-mTORC1 signaling, converging on TFE3/TFEB transcription factors, which in turn regulate expression of Fam134b, a receptor for selective autophagic elimination of the endoplasmic reticulum (ER-phagy). The manuscript is well written, methodologically sounds and addresses important general question of mechanisms underlying ER-phagy regulation. However, I have several concerns specified below and without addressing them I feel that the manuscript is overstated and over-generalized.

(Response) We thank the reviewer for the very important suggestions. We have done our best to address all the comments.

Major points:

1) The authors clearly and convincingly demonstrated that in RCS chondrosarcoma cell line FGF18 stimulates TFEB/TFE3-mediated induction of Fam134b expression. This is a very interesting, novel and, to my knowledge, previously undescribed connection. However, generality of this observation is limited to RCS chondrosarcoma cell line and confirmation in other cell lines or in vivo would add strength and depth to this observation. I would suggest to add MEFs and Saos2 (used by the authors in their recent study EMBO J 2019, 38:e99847) for verification of this connection. This would allow to make more generalized conclusion. Alternatively, in vivo experiments can be expanded (please see my comment below to the presented in vivo data).

(Response) We are glad that the reviewer recognized that our data clearly and convincingly show that FGF18 stimulates TFEB/TFE3-mediated induction of Fam134b expression.

As reported in the response to reviewers 1 and 2, to expand the generalization of this observation we performed both *in vivo* and *in vitro* experiments. We have now demonstrated that starvation-mediated *FAM134B* upregulation, as well as ER-phagy, is impaired in Hela cells lacking TFEB and TFE3 (see Rebuttal fig 4) (in the manuscript: Supplementary Fig. 8f,g). Furthermore, TFEB overexpression induces ER-phagy in U2-OS cells, but not in U2OS in which *FAM134B* was stably silenced (see Rebuttal fig 5) (in the manuscript: Supplementary Fig. 9a,d). In addition, the overexpression of a constitutively active form of TFEB induces ER-phagy in WT, but not in *FAM134B* KO MEFs (see Rebuttal fig 6) (in the manuscript: Supplementary Fig. 9e,f). Taken together these gain- and loss-of-function experiments demonstrated that TFEB/TFE3 activation induces ER-phagy via *FAM134B* transcriptional induction in multiple cell lines.

The description of the new *in vivo* data is included below (point 4).

2) The authors showed that regulation of TFEB/TFE3 by FGF18 is mediated via JNK-dependent degradation of IRS1 leading to down-regulation of AKT-mTORC1 pathway. FGF18-dependent activation of JNK is known and particularly for RCS cells has been demonstrated by these same authors previously (Cinque L et al., Nature 2015, 528:272-276). In relation to AKT-mTORC1 pathway, previously the authors claimed that there is no regulation of Akt/mTORC1 by FGF18 in RCS cells (Suppl Fig 8, Nature 2015, 528:272-276), whereas in the current manuscript they showed down-regulation of Akt/mTORC1 activity by FGF18 (Fig. 4c,d). This clear discrepancy requires clarification.

(Response) The Extended data 8d-e of Cinque et al. Nature 2015 showed the pathway analysis in

the growth plate of FGF18^{+/-} mice, not in RCS cells. Growth plate specimens from FGF18^{+/-} mice showed a reduction of JNK1/2, but not of the Akt/mTORC1 signalling pathway, compared with control. The lack of Akt/mTORC1 signaling defects might be due to the fact we were analysing heterozygous mice, which still express FGF18 from one allele (the KO is embryonic/perinatal lethal). To overcome this limitation, the revised version includes novel data on the generation and analysis of growth plates from FGFR3/4dKO mice demonstrating a significant increase in mTORC1/AKT signalling (Rebuttal fig. 16) (in the manuscript: Fig. 4d), consistent with a role of FGF as negative regulator of the PI3K signalling in cartilage.

Rebuttal figure 16

3) The authors claimed that FGF18 specifically regulates ER-phagy in RCS cells via TFEB/TFE3-Fab134b axis. TFEB/TFE3 are master regulators of autophagy and lysosomal biogenesis in general. Additionally, in the current manuscript (Fig. 1b,c,d,h) and previously (Cinque L et al., Nature 2015, 528:272-276), the authors showed that FGF18 increases general levels of autophagy. Thus, since ER-phagy is a sub-type of the autophagy process, the observed increase in ER-phagy can reflect overall increase in the autophagy and lysosomal biogenesis upon FGF18 stimulation, not selective ER-phagy. To claim selective regulation of ER-phagy by FGF18, changes in the ratio of ER-phagy to autophagy levels should be demonstrated, e.g., EATR assay normalized per LC3-puncta or the number of lysosomes.

(Response) To address this important question we analysed FGF18 mediated autophagy induction in control and TFEB/TFE3 DKO cells. Notably, FGF18 still induces autophagosome biogenesis (Rebuttal fig 17a, manuscript fig. 7a) and autophagosome/lysosome fusion (Rebuttal fig 17b, manuscript fig 4i) in TFEB/TFE3 DKO cells, consistent with the fact that FGF18 stimulation can promote autophagy through multiple mechanisms (e.g. mTORC1 inhibition, Beclin-1 activation etc.). However, the lack of TFEB/TFE3 strongly suppresses CLIMP63 puncta accumulation within autophagosomes and lysosomes (Rebuttal fig 17c and d, manuscript Fig 4f-g). These data clearly show that TFEB/TFE3 activation by FGF18 is essential to activate ER-phagy rather than autophagy in general. We discussed this important observation in the revised version of the manuscript.

Rebuttal figure 17

4) To explore the physiological role of *Fam134b* the authors did morpholino ablation of this gene in medaka fish and claimed its physiological role in endochondral ossification (Fig 2i,j and page 4, paragraph 1). First, there is no endochondral ossification in cranial skeleton of medaka. Bones are formed intra-membranously around cartilaginous skeletal elements, like mandibular bone around Meckel's cartilage and no claims about chondrocyte differentiation can be made. Second, the data are very preliminary, levels of *Fam134b*, lysosomes and CLIMP-63 puncta have to be demonstrated in medaka's chondrocytes with/without *Fam134b* ablation as well as morphological and transcriptional changes in chondrocytes to claim physiological role of *Fam134b* in chondrogenesis.

(Response) We corrected the mistake relative to the ossification in Medaka and performed new experiments to address reviewer requests:

- 1) We demonstrated that FGF18 administration to Medaka fish increases the transcriptional level of *Fam134b*, as well as of the lysosomal protein *Lamp1* (Rebuttal fig 18a) (in the manuscript: Fig. 6c and and supplementary Fig. 10).
- 2) We showed that FGF18 triggers FAM134B-dependent ER-phagy in Medaka cartilage, as demonstrated by colocalization of protein disulfide-isomerase (PDI) in LAMP1-positive

vesicles in control, but not in moFAM134B medaka (Rebuttal fig 18b and c) (manuscript: Fig. 6d,e). We used PDI antibody instead of Climp63, for technical reasons as per compatibility of secondary antibodies (PDI, mouse; Lamp1, Rabbit).

3) We demonstrated, by electron microscopy, that *Fam134b* silencing affects ER homeostasis in chondrocytes of the Medaka, as observed by presence of dilated ER cisternae (Rebuttal fig. 18d). Type II procollagen also accumulates in moFAM134B fish compared to control (Rebuttal fig. 18e) (manuscript: Fig. 6 f,g).

Taken together these data support a physiological role for FGF18-induced ER-phagy via FAM134B in chondrocytes during skeletal development in Medaka.

Rebuttal figure 18

5) Another important point is related to the physiological role of Fam134b – neither mice deficient in Fam134b nor patients with mutated Fam134b were reported signs of growth retardation. This should be discussed.

(Response) We agree with the reviewer. We think that the lack of a clear bone phenotype in FAM134B KO mice, might be probably due to compensatory mechanisms exerted by the other FAM134 family members, namely FAM134A and FAM134C. Indeed, cells lacking FAM134B showed significant upregulation of FAM134A and FAM134C (data not shown). This was the main reason why we decided to analyse FAM134B roles in cartilage using an *in vivo* model with an acute, rather than constitutive, inhibition of FAM134B.

6) In light of the comment above the title is not appropriate. There is no data supporting chondrocyte differentiation. Mentioning chondrocytes in the title is also overstating since only chondrosarcoma cell line was used.

(Response) We agree with the reviewer, our manuscript focused on RCS response to FGF stimulation and not a study on the process of chondrocyte differentiation. However, given that the new version of the manuscript contains several new data in chondrocytes *in vivo* in both medaka and mice, and in different cell lines, we decided to use the following title: " MiT/TFE factors control ER-phagy via transcriptional regulation of FAM134B ".

Minor comments:

Throughout the manuscript the authors made an impression that FGF18 is signaling exclusively via FGFR3 and FGFR4, whereas FGFR1 and FGFR2 are not mediating this signaling. This can be concluded only if the expression of FGFR1 and/or FGFR2 is demonstrated in this particular cell line. The alternative is that RCS cells do not express FGFR1 and FGFR2.

(Response) RCS expresses all four FGF receptors (see Supp fig 1e), however by Crispr-Cas9 single and combined receptor deletion, we demonstrated that FGF18 induced lysosome biogenesis and ER-phagy exclusively through FGFR3 and FGFR4 (see Fig. 1h-j; and Supp fig 1d,f).

In Fig. 3C only TFE3 is significantly upregulated whereas in the text the authors say that TFE3, TFEB and MITF are significantly upregulated. This should be corrected.

(Response) We corrected the figure, which now shows significant effects also in TFEB and MITF expressing samples.

Numerous Western blots lack quantification and it is unclear how reliable are those observations.

(Response) Each blot was representative of at least three independent replica experiments. We included quantification of the data in the revised version of the manuscript.

FGF18 downregulated ribosomal categories by mass-spec, but upregulated their gene expression (Fig. 1a and Fig. S3c). It would be good to discuss if this is a sign of selective elimination of ribosomes by ER-phagy or methodological discrepancy.

(Response) Indeed, this is a very interesting finding. Mass spectrometry experiments demonstrated that FGF18 downregulated ribosomes in WT but not in FAM134B-LIR cells, suggesting that the downregulation of ribosomes is a consequence of ER-phagy. We discussed this aspect in the paper.

Reviewers' comments

Reviewer #1 (Remarks to the Author):

The authors have done a lot of work to revise this paper. However, a number of key concerns remain.

1. I am still worried about the conceptual novelty – we know that TFEB induces autophagy, that FAM134B is important for ER-phagy and collagen homeostasis in relevant cells types, and that FGFs induce autophagy in such cells regulate collagen homeostasis. Settembre's group showed that the effects of FGFs in this context were mediated via JNK and the links between JNK and IRS1 degradation-mTORC1 signalling are well known, as is the regulation of mTORC1 by TFEB. The regulation of FAM134 and ER-phagy by starvation which also activates TFEB has been described before. So this paper looks like it is tying together existing literature.

2. Are the effects of FGF signalling and the proposed downstream mediator on bulk autophagy or is there truly a selectivity conferred by the induction of FAM134B? (Referee 3 point 3). This is important as the FGF and TFEB axes are not novel by themselves as they are known autophagy inducers and the group have previously described how FGF can stimulate autophagy and regulate collagen homeostasis. To address this the authors have tried to show that autophagy is not affected in the TFEB/TFE3 knockout cells – but this goes against much of the previous literature from this lab showing that TFEB is a master transcriptional regulator of autophagy. I am concerned that the key data for this are in Fig 4i and Supp 7a. Supp 7a may be misleading as it does not assess flux and number of vesicles may simply increase due to impaired lysosomal function in the

knockout cells. Fig 4i uses an assay that may be risky as they use FACS and do not count vesicles – I would not trust the FACS assay here at all – the numbers of red-only and yellow vesicles need to be assessed per cell by microscopy. FACS will not assess vesicles and will measure total cellular fluorescence. The authors need to consider that the flux that needs to be assessed is essentially the number of red only vesicles per cell. The ratio is not that informative for the question being addressed – what matters here is the amount of substrate being delivered to lysosomes for degradation per unit time. This is a very important question and even if the assay were performed properly by microscopy (not FACS), I would ask for the authors to assess clearance of a non-ER autophagy substrate in the wild-type and TFEB/TFE3 knockout cells in response to FGF. The mRFP-GFP-LC3 assay along with analyses of one or two non-ER autophagy substrates would allow one to compare with the ER-phagy data and make an assessment whether there was a selective component in this pathway regulates by FGF and TFEB, which is key for this study.

3. Referee 1 comment 2. The degradation kinetics of IRS are not obviously affected by the JNK inhibitor – about 50% is lost with/without the inhibitor compared to the starting amount in graph in rebuttal Fig 1. But graph in Rebuttal fig. 1 and Supplementary fig 2b are different - see JNK inhibitor lanes. Which one is correct?

4. What are the p values in the knockout cells in 4 d and h using t tests? We need to understand if these are true negatives or not. I think t tests are appropriate here (as opposed to ANOVA) in order to make the negative statement in the knockout cells –when I do this for the knockout cells in 4d I get highly significant differences with the FGF treatment. Thus, to my eyes, the raw data in Figs 4d and 4e suggest that FGF very clearly induces FAM134B in TFEB/TFE3 knockout cells, arguing against the mechanism that is being proposed. Note that the graph in 4e looks wrong – the FGF induces FAM134B quite clearly in both wild-type and knockout cells on the blot.

5. I could not find the raw data for Fig 4e or 4h and this may be missing for other figures too.

6. Another key problem pointed out by reviewer 3 is that patients with loss of FAM13B or knockout mice do not have collagen-related phenotypes. The authors suggest that there may be redundancy with other FAM134 family members that accounts for this. But Fig 3b shows negligible effects of FGF on FAM134a or FAM134c. I am also not aware if FAM134a or FAM134c have been shown to regulate ER-phagy. So this point raises concerns about the physiological relevance of the proposed pathway.

7. The authors try to address point 6 by Medaka experiments. The point 4 of reviewer 3 raised concerns about the validity of this model to some aspects of human bone biology. It also made me worry that the authors have only used 1 morpholino to test their hypothesis and I am aware that there has been much debate about the use of morpholinos in fish and how best to avoid off-target effects. I would have thought that at least two oligonucleotides are required – I have checked this now with an expert who has confirmed my concern and sent me this reference. Ideally, one needs two different morpholinos and a rescue experiment. The control and p53 morpholinos are not sufficient to control for off-target effects. Apologies for not noticing this important problem earlier. Key references addressing these issues are:

<https://journals.plos.org/plosgenetics/article?id=10.1371/journal.pgen.1007000>

<https://dev.biologists.org/content/135/10/1735.long>

Reviewer #2 (Remarks to the Author):

The importance of this work remains high and merits publication, as described in my previous comments. Largely, the authors have addressed all my comments satisfactorily. Overall quality of the manuscript remains high.

I have only minor suggestions to be addressed before publication.

Minor points

1) The data presented in response to Reviewer 1 and Reviewer 2 in respect of the relative regulation and function of the two FAM134B variants downstream of FGF18 should be included in the manuscript dataset (nothing to be merely presented to the Reviewers in the rebuttal.) I do think these data are an important corroboration of the role of both FAM134B isoforms in ER-phagy; the authors' work builds upon the premise that this prior observation is wholly correct.

2) Reviewer 3, in point 3, raised an interesting philosophical point regarding measurement of selective autophagy regulation. If upstream stimuli also increase general autophagy, how does one determine whether upregulation of given selective autophagy molecule (e.g. FAM134B), within this response, is quantitatively important in observed upregulation of a given selective autophagy pathway, (e.g. ER-phagy)?

Firstly, it is important to stress that the data from the "EATR" style assay (i.e. Fig. 3e, EATR a.k.a. RAMP4-GFP-RFP), already showed that ER-phagy flux is indeed increased in response to FGF18 and that this depends upon the ER-phagy (LC3-binding) activity of FAM134B.

However, in response to this point of Reviewer 3 the authors now appear to have managed to dissect ER-phagy upregulation from general autophagy upregulation, providing evidence that TFEB knockout cells have no defect in general autophagy but have defective ER-phagy; they have done this, as requested, partly by extending the above EATR/RAMP4-GFP-RFP assays (Fig. 4h,i). This works, as long as one assumes that ER-phagy flux induced by FGF-18 is only a small overall proportion of total autophagy flux (as is likely the case). I am, however, surprised at the complete lack of effect of TFEB knockout on general autophagy. Provision of data corroborating this effect via complementary assays, such as biochemical measurement of lipidated LC3 or SQSTM1 flux, could consolidate this finding and give further confidence in these new data.

Reviewer #3 (Remarks to the Author):

I am very impressed by the extensive and thorough revision performed in response to my suggestions as well as other reviewers. From my point of view, the manuscript matured, acquired depth, strength and generality anticipated for NCB level.

I am very satisfied with the experiments and clarifications in relation to my concerns and have one remaining comment related to my previous concern #5 and corresponding authors' response.

Please add the data for FAM134A and FAM134C into the manuscript (mentioned as "data not shown" in the rebuttal letter) and please mention the lack of phenotype in FAM134B KO mice and potential compensatory mechanisms in the discussion. I believe it is important for the readers to get a more complete picture of these regulatory processes in order not to overestimate the role of FAM134B.

Thanks to the authors for making an exciting story.

Authors' Response - Round 2

Reviewer #1 (Remarks to the Author):

The authors have done a lot of work to revise this paper. However, a number of key concerns remain. 1. I am still worried about the conceptual novelty – we know that TFEB induces autophagy, that FAM134B is important for

ER-phagy and collagen homeostasis in relevant cells types, and that FGFs induce autophagy in such cells regulate collagen homeostasis. Settembre's group showed that the effects of FGFs in this context were mediated via JNK and the links between JNK and IRS1 degradation-mTORC1 signalling are well known, as is the regulation of mTORC1 by TFEB. The regulation of FAM134 and ER-phagy by starvation which also activates TFEB has been described before. So this paper looks like it is tying together existing literature.

(Response) While it is true that the role of FGF, TFEB and mTORC1 kinase in autophagy regulation, as well as of FAM134B in ER-phagy, were known, the mechanisms underlying the hierarchical participation of all of these molecules into a single pathway, which is activated by FGF signaling, during skeletal development is entirely new. We meticulously and systematically characterized an entire and totally novel pathway from the upstream receptors at the plasma membrane to the downstream intracellular proteins and nuclear targets. Notably, our work shows for the first time that FGF signaling controls TFEB/TFE3 factors through an unanticipated regulation of Insulin signaling and that FAM134B is a direct TFEB target, in multiple cell lines and conditions. Hence, we showed that TFEB is not only involved in bulk autophagy, as already described, but it is also involved in the regulation of selective forms of autophagy.

2. Are the effects of FGF signalling and the proposed downstream mediator on bulk autophagy or is there truly a selectivity conferred by the induction of FAM134B? (Referee 3 point 3). This is important as the FGF and TFEB axes are not novel by themselves as they are known autophagy inducers and the group have previously described how FGF can stimulate autophagy and regulate collagen homeostasis. To address this the authors have tried to show that autophagy is not affected in the TFEB/TFE3 knockout cells – but this goes against much of the previous literature from this lab showing that TFEB is a master transcriptional regulator of autophagy. I am concerned that the key data for this are in Fig 4i and Supp 7a. Supp 7a may be misleading as it does not assess flux and number of vesicles may simply increase due to impaired lysosomal function in the knockout cells. Fig 4i uses an assay that may be risky as they use FACS and do not count vesicles – I would not trust the FACS assay here at all – the numbers of red-only and yellow vesicles need to be assessed per cell by microscopy. FACS will not assess vesicles and will measure total cellular fluorescence. The authors need to consider that the flux that needs to be assessed is essentially the number of red only vesicles per cell. The ratio is not that informative for the question being addressed – what matters here is the amount of substrate being delivered to lysosomes for degradation per unit time. This is a very important question and even if the assay were performed properly by microscopy (not FACS), I would ask for the authors to assess clearance of a non-ER autophagy substrate in the wild-type and TFEB/TFE3 knockout cells in response to FGF. The mRFP-GFP-LC3 assay along with analyses of one or two non-ER autophagy substrates would allow one to compare with the ER-phagy data and make an assessment whether there was a selective component in this pathway regulates by FGF and TFEB, which is key for this study.

(Response) TFEB/TFE3 dKO cells showed down-regulation of TFEB-target genes transcription, including FAM134B, and reduced lysosomal function (suppl fig 5e). However, keeping in mind that the FGF is quite upstream in the signaling cascade it could still induce autophagy through multiple independent mechanisms, which do not necessarily involve TFEB activity, such as Beclin1 complex regulation (Cinque et al. 2015), inhibition of mTORC1 and AKT (this work), which controls autophagy through multiple downstream effectors. Indeed, this is not surprising since there are several "TFEB-independent" autophagy regulation mechanisms. As suggested by reviewer 2, we performed quantification of the red-only vesicles by mCherry-GFP-LC3 tandem experiment and biochemical measurement of lipidated LC3 to consolidate this finding and give further confidence in these new data (**Rebuttal Figure 1**). The data clearly show that FGF still induce autophagosome biogenesis and autophagosome-lysosome fusion in TFEB/TFE3 knockouts cells.

FIGURE 1

Figure 1: a. Representative immunofluorescence staining of mCherry-GFP-LC3 Tandem experiment in RCS with indicated genotypes, treated with vehicle (5% ABS) and FGF18 (50 ng/mL) for 12 hours. Nuclei were stained with DAPI (blue). **b.** Bar graph shows quantification of mCherry⁺ vesicles/cell (autolysosomes) and mCherry⁺/GFP⁺ vesicles/cell (autophagosomes). Mean +/- standard error of the mean (sem) of N=24 (veh), N=30 (FGF18), N=27 (TFEB;3KO veh), N= 33 (TFEB;3KO FGF18) cells. Student un-paired T-Test ***p<0.0005; **p<0.005. **c.** Western blot analysis of LC3 in RCS chondrocytes with indicated genotypes, treated with vehicle (5% ABS) and FGF18 (50 ng/mL) for 12h. BafA1 (200 nM; 4 hours) was used to inhibit lysosome activity. β -actin was used as loading control. Blots are representative of N=5 independent experiments.

3. Referee 1 comment 2. The degradation kinetics of IRS are not obviously affected by the JNK inhibitor – about 50% is lost with/without the inhibitor compared to the starting amount in graph in rebuttal Fig 1. But graph in Rebuttal fig. 1 and Supplementary fig 2b are different - see JNK inhibitor lanes. Which one is correct?

(Response) The different graphs in Rebuttal fig.1 and supplementary figure 2b depend on the fact that additional experiments were performed in Supplementary and final fig2b. Of note, in all experiments performed we clearly showed that JNK inhibitor affects IRS1 phosphorylation at all time points of FGF18 treatment, whereas at later time points induces an increase of total IRS1 levels (at 6h and 8h of FGF18 treatment + JNK inhibitor compared to only FGF18-treated cells).

4. What are the p values in the knockout cells in 4 d and h using t tests? We need to understand if these are true negatives or not. I think t tests are appropriate here (as opposed to ANOVA) in order to make the negative statement in the knockout cells – when I do this for the knockout cells in 4d I get highly significant differences with the FGF treatment. Thus, to my eyes, the raw data in Figs 4d and 4e suggest that FGF very clearly induces FAM134B in TFEB/TFE3 knockout cells, arguing against the mechanism that is being proposed. Note that the graph in 4e looks wrong – the FGF induces FAM134B quite clearly in both wild-type and knockout cells on the blot.

(Response) Figure 4d shows 8.5-fold increase in Fam134b mRNA levels in control cells treated with FGF18 compared to 2.4-fold increase in TFEB/TFE3dko cells. Even using a different statistical analysis, these differences remains highly significant. Similarly, the blot figure 4e clearly shows a very strong impairment of FAM134B induction at protein level in TFEB/TFE3dko cells compared to control. The reviewer also argues that a different statistical test should have been used to analyze the data in figure 4e. The t-test compares the means between 2 samples, but if there are more than 2 conditions in an experiment an ANOVA is required. In these experiments we have 4 groups of samples, and by one-way ANOVA analysis we took in account overall the significance of the experiment set, the variability internally the groups of samples and the variability between the groups. Hence, ANOVA analysis makes stronger the significance of the results.

5. I could not find the raw data for Fig 4e or 4h and this may be missing for other figures too.

(Response) The data are in the excel file named Source data Cinque et. Al.

6. Another key problem pointed out by reviewer 3 is that patients with loss of FAM13B or knockout mice do not have collagen-related phenotypes. The authors suggest that there may be redundancy with other FAM134 family members that accounts for this. But Fig 3b shows negligible effects of FGF on FAM134a or FAM134c. I am also not aware if FAM134a or FAM134c have been shown to regulate ER-phagy. So this point raises concerns about the physiological relevance of the proposed pathway.

(Response) Reviewer 3 asked to comment the observations that FAM134B mutations in patients and mice do not seem to lead to a major skeletal phenotype. To our knowledge in depth studies in mice and humans have not been performed yet, so we have no clue whether they present skeletal alterations or not. We commented that the lack of phenotype could be due to the compensation by the other FAM134 family members, given that other Fam134 proteins harbor a LIR and reticulon homology domains, interact with LC3 proteins, and are therefore likely to be functionally equivalent to FAM134B. Indeed a recent work demonstrated a role of FAM134A in ER-phagy regulation (Liang et al. Cell 2020). This compensation mechanism might occur when FAM134B loss-of-function mutations are present, independently of FGF.

7. The authors try to address point 6 by Medaka experiments. The point 4 of reviewer 3 raised concerns about the validity of this model to some aspects of human bone biology. It also made me worry that the authors have only used 1 morpholino to test their hypothesis and I am aware that there has been much debate about the use of morpholinos in fish and how best to avoid off-target effects. I would have thought that at least two oligonucleotides are required – I have checked this now with an expert who has confirmed my concern and sent me this reference. Ideally, one needs two different morpholinos and a rescue experiment. The control and p53 morpholinos are not sufficient to control for off-target effects. Apologies for not noticing this important problem earlier. Key references addressing these issues are:

<https://journals.plos.org/plosgenetics/article?id=10.1371/journal.pgen.1007000>

<https://dev.biologists.org/content/135/10/1735.long>

(Response) Fish models represent a suitable *in vivo* model for the study of skeletal development and several skeletal dysplasia are studied using fish as model for Osteogenesis Imperfecta (Imke AK et al, J Bone Miner Res 2018; Gioia R, Hum Mol Genet 2017), Osteopetrosis (Thuy Than et al, Comp Biochem Physiol 2015) or Osteoporosis (Thuy Than et al, Development 2012), to cite some. As per reviewer 3 request, we performed new additional experiments in Medaka fish to demonstrate the physiological relevance of our pathway. Reviewer 3 was fully satisfied with our experiments.

As per reviewer request, we have performed rescue experiments. We injected an mRNA encoding for the human version of FAM134B tagged with HA. We verified the expression of the FAM134B-HA protein and observed complete rescue of both size and ossification of the Morphant fish demonstrating that the phenotype of MoFAM134B fish was due to the lack of FAM134B (**Rebuttal Figure 2**).

FIGURE 2

Figure 2: a. Western blot analysis of HA-tag from a pool of medaka fish embryos injected with scramble or injected with

mRNA produced from human HA-FAM134B pcdna3.1(+). β -actin was used as loading control. **b.** Bar graphs show quantification of total length and head size of medaka fish model of *Fam134b^{mo}* and mRNA-injected *Fam134b^{mo}* expressed as % relative to the scramble. Mean \pm standard error of the mean (sem) of at least n=8 fish per genotype. Student un-paired T-Test * $p < 0.05$; ns, not significant. **c,d.** Bar graphs show quantification of Alcian blue (cartilage) (c) and Alizarin Red (bone) (d) staining of *Fam134b^{mo}* and mRNA-injected *Fam134b^{mo}*. Ethmoid plate (EP), Palatoquadrate (PQ), Ceratohyal (CH), Paired Prootics (PO), Ceretobranchials 1 to 5 (CB1 to CB5) cartilage length (c) and bone mineralization (d) were evaluated. Values were expressed as % relative to the scramble (100% red dotted line). Mean \pm standard error of the mean (sem) of at least n=6 fish/genotype. Student un-paired T-Test * $p < 0.05$; ** $p < 0.005$; *** $p < 0.0005$ for comparison between *Fam134b^{mo}* and mRNA-injected *Fam134b^{mo}* Medaka.

Reviewer #2 (Remarks to the Author):

The importance of this work remains high and merits publication, as described in my previous comments. Largely, the authors have addressed all my comments satisfactorily. Overall quality of the manuscript remains high.

I have only minor suggestions to be addressed before publication.

Minor points

1) The data presented in response to Reviewer 1 and Reviewer 2 in respect of the relative regulation and function of the two FAM134B variants downstream of FGF18 should be included in the manuscript dataset (nothing to be merely presented to the Reviewers in the rebuttal.) I do think these data are an important corroboration of the role of both FAM134B isoforms in ER-phagy; the authors' work builds upon the premise that this prior observation is wholly correct.

(Response) The data have now been included in the manuscript (Supplementary figure 7d,e).

2) Reviewer 3, in point 3, raised an interesting philosophical point regarding measurement of selective autophagy regulation. If upstream stimuli also increase general autophagy, how does one determine whether upregulation of given selective autophagy molecule (e.g. FAM134B), within this response, is quantitatively important in observed upregulation of a given selective autophagy pathway, (e.g. ER-phagy)?

Firstly, it is important to stress that the data from the "EATR" style assay (i.e. Fig. 3e, EATR a.k.a. RAMP4-GFP-RFP), already showed that ER-phagy flux is indeed increased in response to FGF18 and that this depends upon the ER-phagy (LC3-binding) activity of FAM134B. However, in response to this point of Reviewer 3 the authors now appear to have managed to dissect ER-phagy upregulation from general autophagy upregulation, providing evidence that TFEB knockout cells have no defect in general autophagy but have defective ER-phagy; they have done this, as requested, partly by extending the above EATR/RAMP4-GFP-RFP assays (Fig. 4h,i). This works, as long as one assumes that ER-phagy flux induced by FGF-18 is only a small overall proportion of total autophagy flux (as is likely the case). I am, however, surprised at the complete lack of effect of TFEB knockout on general autophagy. Provision of data corroborating this effect via complementary assays, such as biochemical measurement of lipidated LC3 or SQSTM1 flux, could consolidate this finding and give further confidence in these new data.

(Response) TFEB/TFE3 dKO cells showed down-regulation of TFEB-target genes transcription and reduced lysosomal function (suppl fig 5e). However, keeping in mind that the FGF is quite upstream in the signaling cascade it could still induce autophagy through multiple independent mechanisms, which do not necessarily involve TFEB activity, such as Beclin1 complex regulation (Cinque et al. 2015), inhibition of mTORC1 and AKT (this work), which control autophagy through multiple downstream effectors. Since the discovery of TFEB, we made similar observations multiple times, as there are several "TFEB-independent" autophagy regulation mechanisms. As suggested by reviewer 2, we performed quantification of the red-only vesicles by mCherry-GFP-LC3 tandem experiment and biochemical measurement of lipidated LC3 to consolidate this finding and give further confidence in these new data. The data clearly show that FGF still induce autophagosome biogenesis and autophagosome-lysosome fusion in TFEB/TFE3 knockouts cells (see **Rebuttal Figure 1**).

Reviewer #3 (Remarks to the Author):

I am very impressed by the extensive and thorough revision performed in response to my suggestions as well as other reviewers. From my point of view, the manuscript matured, acquired depth, strength and generality anticipated for NCB level.

I am very satisfied with the experiments and clarifications in relation to my concerns and have one remaining comment related to my previous concern #5 and corresponding authors' response.

Please add the data for FAM134A and FAM134C into the manuscript (mentioned as "data not shown" in the rebuttal letter) and please mention the lack of phenotype in FAM134B KO mice and potential compensatory mechanisms in the discussion. I believe it is important for the readers to get a more complete picture of these regulatory processes in order not to overestimate the role of FAM134B.

We have currently an ongoing project on the role of other FAM134 family members in the regulation of ER-phagy and for this reason we kindly ask to not include these data in the manuscript.

Thanks to the authors for making an exciting story.

Thank you for submitting your manuscript entitled "MiT/TFE factors control ER-phagy via transcriptional regulation of FAM134B" [EMBOJ-2020-105696] to The EMBO Journal. I have discussed it and the existing referees' reports with the editorial team and also sought advice from a good expert in the field.

The external advisor concurs with us on the general interest of your findings and supports publication of the study in its present form. Given this opinion from a trusted expert in the field, we are offering to pursue publication of your manuscript here, conditioned to text editing and clarification of some editorial issues.

Our external advisor states "I agree that more research is needed to determine whether the observed ER-phagy is part of the general autophagy pathway, however this in my mind should be the subject of future studies" Therefore, I would ask you to discuss this point in the manuscript.

We are now performing routine pre-acceptance checks on the manuscript and I will get back to you shortly with a list of points that need to be addressed before we can formally accept your work.

Thank you for giving us the chance to consider your manuscript for publication in The EMBO Journal. Please feel free to contact me should you have any further questions.

Authors' Response to The EMBO Journal**13th Jun 2020**

Please find below a point by point response to the requests that were listed in your previous communications.

external advisor-> I agree that more research is needed to determine whether the observed ER-phagy is part of the general autophagy pathway, however this in my mind should be the subject of future studies"

We have included the following statement in the discussion section: "Future studies will be needed to investigate whether additional autophagy substrates are delivered to lysosomes through similar mechanisms, or whether ER-phagy is part of the general autophagy pathway".

2nd Editorial Decision - The EMBO Journal**18th Jun 2020**

I am pleased to inform you that your manuscript has been accepted for publication in The EMBO Journal.

Congratulations!

Corresponding Author Name: Carmine Settembre

Manuscript Number: